# The model diatom *Phaeodactylum tricornutum* provides insights into the diversity and function of microeukaryotic DNA methyltransferases

Antoine Hoguin[1,3], Feng Yang[2], Agnès Groisillier [2], Chris Bowler [1], Auguste Genovesio[1], Ouardia Ait-Mohamed [1✉], Fabio Rocha Jimenez Vieira [1,4✉] & Leila Tirichine [2✉]

Cytosine methylation is an important epigenetic mark involved in the transcriptional control of transposable elements in mammals, plants and fungi. The Stramenopiles-Alveolate-Rhizaria (SAR) lineages are a major group of ecologically important marine micro-eukaryotes, including the phytoplankton groups diatoms and dinoflagellates. However, little is known about their DNA methyltransferase diversity. Here, we performed an in-silico analysis of DNA methyltransferases found in marine microeukaryotes and showed that they encode divergent DNMT3, DNMT4, DNMT5 and DNMT6 enzymes. Furthermore, we found three classes of enzymes within the DNMT5 family. Using a CRISPR/Cas9 strategy we demonstrated that the loss of the DNMT5a gene correlates with a global depletion of DNA methylation and overexpression of young transposable elements in the model diatom *Phaeodactylum tricornutum*. The study provides a view of the structure and function of a DNMT family in the SAR supergroup using an attractive model species.

[1] Institut de Biologie de l'Ecole Normale Supérieure (IBENS), Ecole Normale Supérieure, CNRS, INSERM, PSL Université Paris, 75005 Paris, France. [2] Nantes Université, CNRS, US2B, UMR 6286, F-44000 Nantes, France. [3] Present address: Institute of Molecular Plant Sciences, University of Edinburgh, Edinburgh EH9 3BF, UK. [4] Present address: Laboratory of Computational and Quantitative Biology—LCQB - UMR 7238 CNRS—Sorbonne Université. Institut de Biologie Paris Seine, 75005 Paris, France. ✉email: ouardia.mohamed@gmail.com; fabio.rocha_jimenez_vieira@sorbonne-universite.fr; tirichine-l@univ-nantes.fr

In eukaryotes the methylation of the fifth carbon of cytosine (5mC) is a well-known epigenetic mark associated with transcriptional repression. It has been implicated in a wide range of cellular processes including the stability of repeat rich centromeric and telomeric regions as well as in repression of transposable element (TEs) expression[1–4]. 5mC is deposited by DNA methyltransferases (DNMTs) capable of de novo methylation and is propagated through subsequent cell division by maintenance DNMT enzymes. Eukaryotes have acquired a diverse set of DNMTs by horizontal gene transfer of bacterial DNA cytosine methyltransferase involved in the restriction-methylation system[5]. All DNMTs contain a catalytic protein domain composed of ten conserved motifs (annotated I to X) that provide binding affinity to the DNA substrate and the methyl donor cofactor S-Adenosyl methionine to process the transfer of a methyl group to unmethylated cytosines[6,7]. DNMTs have further diversified over evolutionary time scales in eukaryote lineages and acquired chromatin associated recognition and binding domains giving rise to a wide diversity of DNA methylation patterns[8,9].

The loss and gain of DNMTs have been associated with profound divergence in cell biology and control of gene expression. To date, six main eukaryotic DNMT families have been described and named DNMT1, DNMT2, DNMT3, DNMT4, DNMT5, and DNMT6[10,11]. In Metazoans, the combined activity of the DNMT3 and DNMT1 families allow the deposition and maintenance of DNA methylation patterns during the successive developmental waves of DNA demethylation and remethylation[12]. Zebrafish possesses six DNMT3 family de novo methyltransferase genes, dnmt3–dnmt8. This group includes both orthologs of mammalian dnmt3a and dnmt3b as well as fish-specific genes with no mammalian orthologs[13]. In fungi, the DNA methylation machinery consists in a maintenance activity by DNMT1/DIM2, as in Neurospora crassa[14], or the activity of ATPase-DNMT5 enzymes as reported in Cryptococcus neoformans[11,15]. The DNMT5 enzyme also correlates with a heavy histone linker DNA methylation landscape in Micromonas pusilla, the pelagophyte Aureococcus annophagefferens and the haptophyte Emiliania huxleyi[11]. Fungal DNMT4 relatives are involved in the DNA methylation related process known as Repeat-Induced Point Mutation and Methylation Induced Premeiotically that leads to TE extinction and/or stage specific repression as observed in Aspergillus and Neurospora species[16–19].

Losses and lineage specific duplication of DNMT1 and DNMT3 have occurred during insect evolution, such as in Diptera lineages[20], leading to secondary loss of global 5mC methylation. In plants, the acquisition of novel DNMT1 proteins named Chromomethylases (CMTs) and the divergence of the DNMT3 family led to the spreading of the asymmetrical non-CG patterns of DNA methylation that is extensively found in angiosperms[21–23]. DNMT2 is known to methylate tRNAs to yield ribo-5-methylcytidine in a range of eukaryotic organisms, including humans, mice, Arabidopsis thaliana, and Drosophila melanogaster[24]. It is characterized by its cytoplasmic localization that contrasts with the exclusively nuclear localization of Dnmt1 and Dnmt3[25]. Lastly, DNMT6 has been found in Chlorophyta, Haptophyta, Ochrophyta, diatoms and dinoflagellates (e.g., Symbiodinium kawagutii and Symbiodinium minutum)[10,11,26,27] but its function remains elusive. Importantly, 5mC is increasingly reported in eukaryotes of the Stramenopiles-Alveolate-Rhizaria (SAR) lineages as in dinoflagellates[26], diatoms[27], and kelps[28]. However, because of the severe underrepresentation of marine unicellular eukaryotes in modern sequencing databases, our understanding of the DNA methylation machinery in these organisms remains scarce.

Diatoms are a dominant, abundant, and highly diverse group of unicellular brown microalgae (from 2 to 200 μm) of the stramenopile lineage. It is estimated that diatoms are responsible for nearly 20% of primary production on earth[29,30]. They are known to dominate marine polar areas and are major contributors of phytoplankton oceanic blooms. To date, 5mC has been reported only in four diatoms, Thalassiosira pseudonana[11], Cyclotella cryptica[31], Fragilariopsis cylindrus[11], and Phaeodactylum tricornutum[11,27]. Diatom methylation patterns strongly contrast with the patterns observed in animals but also dinoflagellates and plants[32]. Firstly, in P. tricornutum, T. pseudonana and F. cylindrus, total levels of DNA methylation range from 8% to as low as 1% of cytosines in the CG context[11] over repeats and TEs usually (but not exclusively) concentrated in telomeric regions[11,27]. Non-CG methylation is also detected but is scarce. Diatom genomes are therefore predominantly composed of isolated highly CG methylated TE islands in an otherwise unmethylated genome and to that regard are remarkably like fungal methylation profiles. In all diatoms examined so far, methylated TEs often have low expression[11,27,31]. This is remarkably consistent with the repressive role of DNA methylation in other eukaryotes and further traces back 5mC-mediated control of TE expression to the last eukaryotic common ancestor. Nonetheless, direct evidence of 5mC repressive role of TEs in diatoms is lacking. Diatom genomes contain predicted proteins similar to members of the DNMT2, DNMT3, DNMT4, DNMT5, and DNMT6 family[11,33]. The conservation of their domain composition across eukaryotic groups as in the yeast Cryptococcus neoformans suggests that diatom DNMT5-like C5-MTases play a conserved and specific role in DNA methylation[11,15]. However, the functions of the DNMTs reported in diatoms have not been characterized in vivo.

Recent advances in high throughput RNA sequencing technologies led to the development of the Microbial Eukaryote Transcriptome Sequencing Project (MMETSP)[34]. The MMETSP concatenates more than 650 transcriptomes from diverse microeukaryote lineages, such as diatoms and dinoflagellates, making it the biggest sequence database for transcriptomes from individual marine microeukaryotes. Here, utilizing the newly defined enhanced Domain Architecture Framework (eDAF) methodology[35], we first explored the structural and phylogenetic diversity of DNMT sequences in marine microeukaryotes from the publicly available MMETSP sequencing databases. Using an integrative approach with available genomes and phylogenetic studies, we provide a DNMT phylogeny focused on the structural and domain diversity found in microeukaryote enzymes and discuss their evolutionary origins. We define, in the DNMT5 family, the sub-families DNMT5a, b and c enzymes, based on structure and phylogenetic assessment. The presence of the predicted DNMT5 family diversity remarkably contrasts with the apparent lack of DNMT1 in most of the MMETSP and microeukaryote databases. Using CRISPR/Cas9 genome editing, we present the functional characterization of the DNMT5a subfamily in the model diatom P. tricornutum demonstrating, to our knowledge for the first time in any SAR, the role of this family in the repression of TEs in an early diverging eukaryote lineage[36].

## Results

**Diversity of DNMT5 methyltransferases in microeukaryotes.** To capture the diversity of 5-cytosine DNA methyltransferases encoded in microalgae, we applied a relaxed HMMER search ($e = 1$ as the cut-off threshold) for the PFAM DNMT (PF00145) domain on transcriptomes from the MMETSP database. The aim of this approach was to retrieve the maximum of DNMT hits between distantly related eukaryotes. In this study, we focused on DNMT1, DNMT3, DNMT4, DNMT5, and DNMT6 gene families that are known or represent putative DNA modifying enzymes.

We retained sequences showing conserved DNMT domains and depicted their domain structures using eDAF analysis[35]. We built a representative phylogeny of DNA methyltransferases based on the alignment of conserved DNMT motifs (Fig. 1a, Supplementary Fig. 1, Supplementary Data 1). Since DNMT2 is an aspartic acid transfer RNA methyltransferase[25], published microalgal DNMT2 sequences were used as additional sequences for phylogenetic analysis. The tree construction exploited the stability of Bayesian approaches to deal with the fast evolution rates observed in our DNMT sequences. Methods based on posterior probabilities present more support values than random sampling algorithms when facing high mutation rates[37–39]. The MMETSP database is composed of transcripts and is thus deprived from lowly expressed genes or pseudogenes. The relative absence of a given gene family within a species will therefore be interpreted accordingly.

We found three gene families related to the DNMT5 clade of enzymes that we named DNMT5a, DNMT5b, and DNMT5c (Fig. 1a). The sequence alignments show homology in the functional DNMT motifs (I-IV, VII, and X) that contain the S-Adenosyl methionine binding and catalytic domains within DNMT5s (i.e., DNMT5a, b and c) (Supplementary Fig. 2). We noticed that the DNMT5 S-Adenosyl methionine-binding phenylalanine found in the catalytic motif IV of other DNMTs is replaced by a serine. Along with other amino acid substitutions within the motifs I to V (Supplementary Fig. 2), this serine locks the protein in an autoinhibitory state released by hemi-methylated DNA binding and the ATPase activity of the SNF domain[40]. The three DNMT5 families form a supported group of enzymes (posterior probabilities 0.94). The DNMT5a and DNMT5b clades are well supported (posterior probabilities of 0.98 and 0.97, respectively). The DNMT5c family is, however, less supported (posterior probability of 0.88). The relationships between the DNMT5a, b, c sequences are, however, unresolved as the DNMT5a, b branch is poorly supported (posterior probability of 0.51). Of note, DNMT5a is found in distantly related eukaryote lineages. We found 76 species with at least one DNMT5 orthologue. We found a DNMT5a in the green algae *Tetraselmis astigmata* but also in haptophytes and the marine photosynthetic excavate euglenozoa *Eutreptiella gymnastica*. The DNMT5a family is also found in stramenopiles, including diatoms, bolidomonas, pelagophytes, and dictiochophytes, as well as in fungi (former *Cryptococcus* DNMT5-related enzymes) (Fig. 1a, Supplementary Data 2). The DNMT5b enzyme is found in diatoms, *Bolidomonas pacifica* and haptophytes. *Emiliania huxleyi* DNMT5 enzymes are not found in other haptophytes in the MMETSP database. In addition, the phylogenetic position of E. huxleyi DNMT5 proteins are not very supportive. First, E. huxleyi DNMT5a protein is not monophyletic with other ochrophyte DNMT5s and its DNMT5b appears as an outgroup found in diatoms. (Supplementary Fig. 1). Within diatoms, genomes from both *F. cylindrus* and *Synedra* contain DNMT5a and DNMT5b gene copies (Supplementary Data 3) but lineage specific loss of DNMT5a is also observed in Thalassiosirales. Haptophyte DNMT5s could be of lateral gene transfer origin, as in other microalgae. DNMT5c enzymes are specific to dinoflagellates that are known to have very fast evolutionary rates and likely divergent base/amino acid compositions. Dinoflagellate DNMT5c sequences may thus represent a highly divergent DNMT subgroup that our phylogeny failed to associate with other DNMT5s.

We found that the DNMT5a and b families share a C-terminal SNF2-type DEXDc/HELICc helicase domain composed of two helicases complemented or not by a RING finger domain (Fig. 1b, Supplementary Data 4). We found that DNMT5b enzymes display unique features. First, among them, 14 contain an N-terminal laminin B receptor domain as in *T. pseudonana* (Fig. 1b, Supplementary Data 4). Also, other DNMT5b enzymes contain N-terminal CpG methyl binding domains, as well as HAND structure domains, methyl-lysine and methyl-arginine TUDOR binding domains (Supplementary Data 4). Finally, their DNMT domain is longer compared to the DNMT5a, c due to the presence of spacer sequences between motifs. These differences in structure may highlight functional diversity between the DNMT5 subfamilies and are consistent with the duplication followed by divergence hypothesis described above. Accordingly, the DNMT5c family also diverged compared to the DNMT5a and b enzymes at the protein domain composition. It is indeed characterized by a long (~1000 amino-acids) N-terminal sequence with no annotated functional domains (Fig. 1b, Supplementary Data 4).

**The DNMT4 and DNMT1 family methyltransferases in microalgae.** In our phylogenetic analysis, the DNMT4 and DNMT1 clades form a poorly supported gene family, as previously described[11,41] (Fig. 1a, Supplementary Fig. 1). DNMT1s are maintenance enzymes in eukaryotes that often associate a DNMT catalytic domain with chromatin binding domains such as Bromo-Adjacent Homology domains, Plant HomeoDomains, chromodomains and domains required for interaction with accessory proteins. DNMT4 enzymes are related to DIM2 enzymes in fungi[41] and are involved in the Methylation Induced Premeiotically and Repeat-Induced Point Mutation processes. Interestingly, two DNMT4 enzymes were also described in the pennate diatom *F. cylindrus* and the centric diatom *T. pseudonana* based on a previous phylogenetic analysis of DNMT enzymes in microalgae[10]. We first confirmed that orthologues of *T. pseudonana* DNMT4 enzymes are widespread in diatom transcriptomes and genomes. A total of 8 pennate and 23 centric diatoms out of 60 diatom species, express or encode at least one DNMT4 related transcript (Supplementary Data 3). This finding suggests that the family is ancestral in diatoms. In our analysis, no DNMT4 enzymes were found in other species of the MMETSP database. Phylogenetic analysis indicates that RID and diatom DNMT4s may form a moderately supported monophyletic family of enzymes (Fig. 1a). At the structural level, both RID and diatom DNMT4 enzymes diverged compared to DNMT1 enzymes, and also between each other. Most diatom DNMT4 enzymes are composed of a single DNMT domain as in *T. pseudonana*, which also contrasts with fungal enzymes (Fig. 1b, Supplementary Data 4). Nonetheless, nine diatom DNMT4 proteins possess an additional N-terminal chromodomain as observed in *Thalassiosira miniscula* (Fig. 1b, Supplementary Data 3 and 4). Furthermore, we found two putative DNMT1-like enzymes in the transcriptomic database of two *Raphidophyceae* brown microalgae: *Heterosigma akashiwo* and *Chatonella subsalsa*. They are composed of a conserved DNMT domain and a plant homeodomain (Fig. 1b, Supplementary Fig. 1, Supplementary Data 4) but poorly defined a monophyletic gene family with either DNMT1s or DNMT4s.

Interestingly, we found a DNMT1-related enzyme in three haptophyte species out of four (*Gephyrocapsa oceanica*, *Isochrysis.sp-CCMP1324* and *Coccolithus pelagicus*) from the MMETSP database that cluster with annotated CMTs found in the coccolithophore *E. huxleyi* (Fig. 1a, Supplementary Fig. 1). We found that the enzymes of *Gephyrocapsa oceanica* (CAMPEP_0188208858), *Isochrysis-CCMP1324* (CAMPEP_0188844028) and *Emiliania huxleyi* (jgi_215571) have DNMT1-like structures with a Replication Foci Domain followed by a Bromo-Adjacent Homology (in *Emiliana huxleyi* only) and a conserved DNMT domain (Fig. 1b, Supplementary Data 4). Haptophyte enzymes

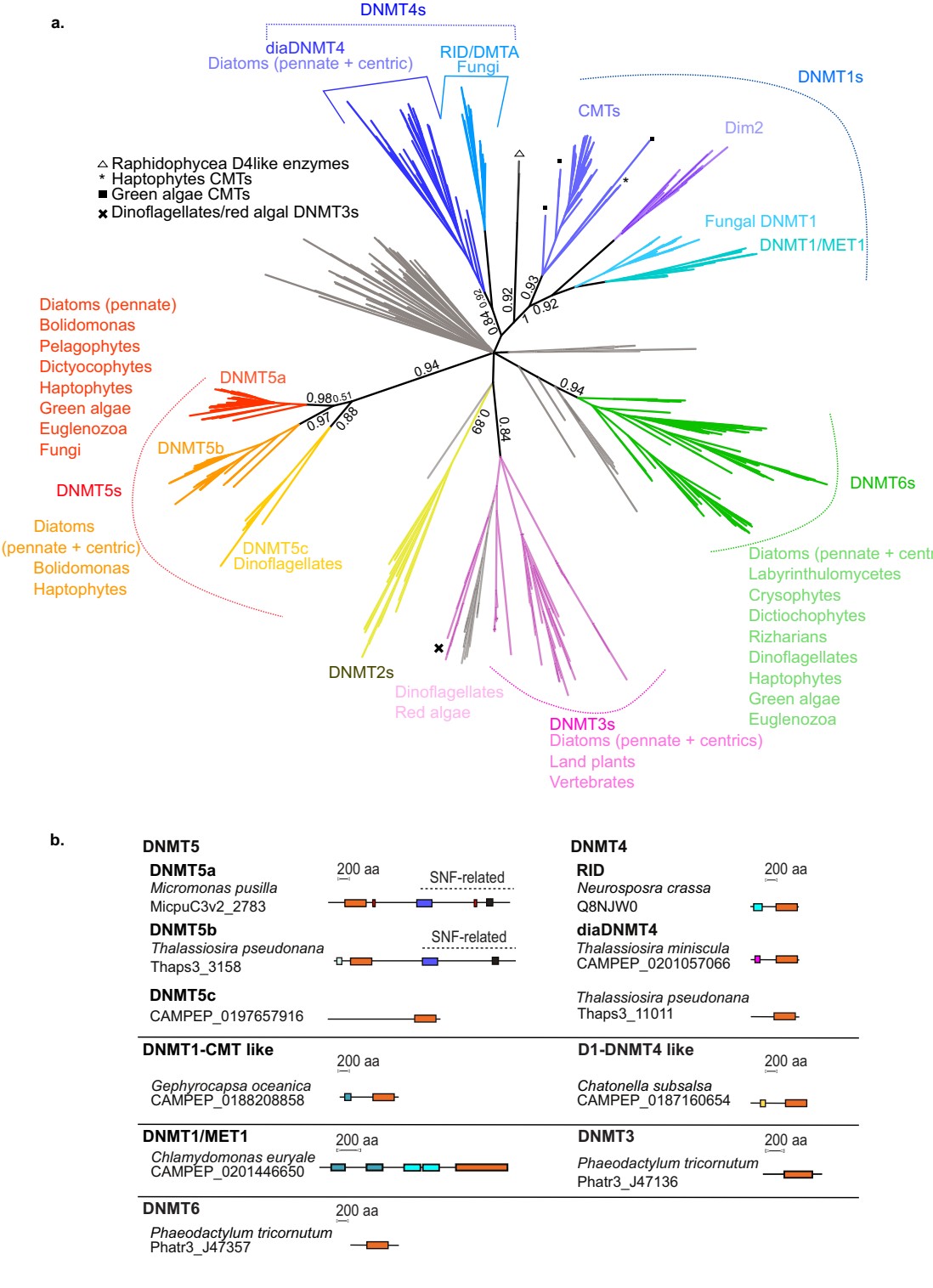

**Fig. 1 Phylogenetic analysis of DNMTs from MMETSP. a** Convergent phylogenetic tree of DNMT domains from the MMETSP and reference genome databases. The sequences selected were from microeukaryotes. Numbers represent MrBAYES posterior probabilities. Gray branches represent bacterial and viral DNA cytosine methyltransferase enzymes. We indicate the main lineages found within each gene family using their corresponding colors next to the tree. **b** Schematic representation of the DAMA/CLADE structure of representative DNMT enzymes. DNMT DNA methyltransferase; RING: Ring zinc finger domain, DX Dead box helicase, Hter C-terminus-Helicase, LBR Laminin B receptor, RFD Replication Foci Domain, BAH Bromo-Adjacent Homology, Chromo Chromodomain, PHD Plant HomeoDomain.

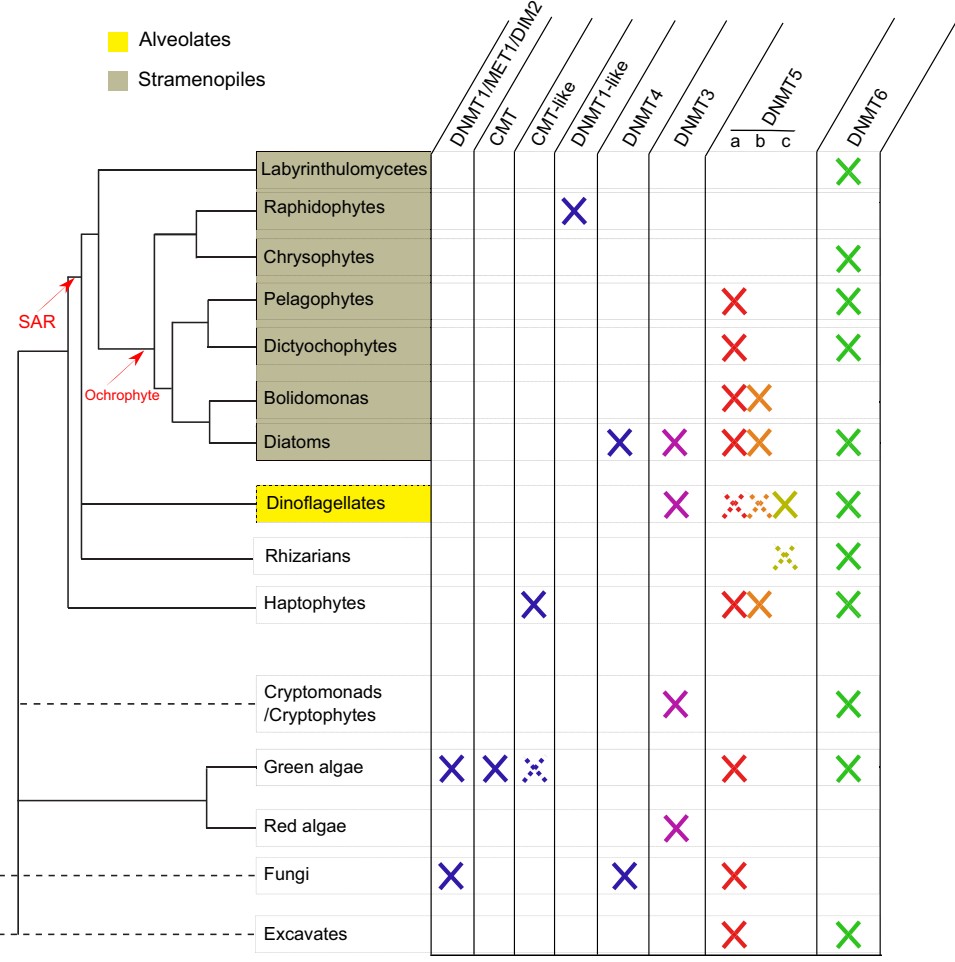

**Fig. 2 Summary of DNMT family lineages found in microeukaryotes.** Full crosses report the presence of a given gene family within lineages. Dashed lines and crosses indicate the uncertainty in the eukaryotic phylogeny as well as low support presence of a given DNMT family within lineages. Fungi that share DNMT families with other eukaryotes presented in this study are shown for comparison purposes. SAR: Stramenopile Alveolate Rhizaria lineage. Ochrophyte are secondary endosymbiont, photosynthetic lineages of stramenopiles.

seem to distantly relate to the conserved green algal CMT (hCMT2) enzymes (Fig. 1a, Supplementary Fig. 1).

We detected DNMT1/MET1 transcripts encoding proteins similar to the plant MET1 enzyme in seven green algae species from MMETSP, mainly from the Chlamydomonadales order such as in some *Chlamydomonas* species (Fig. 1b, Supplementary Fig. 1, Supplementary Data 2).

**The DNMT3 and DNMT6 methyltransferases are abundant in diatoms and lack chromatin associated domains**. Our data indicate that the DNMT3 family is not particularly frequent in microalgae (Fig. 2, Supplementary Data 2). DNMT3 is absent in most stramenopiles except in diatoms; for which genomic and transcriptomic data strongly support its presence (Supplementary Data 3). DNMT3 seems absent in the studied haptophytes (Fig. 2, Supplementary Data 2). Only one transcript from the crypto-monad *Goniomonas pacifica* could be annotated as DNMT3. In addition, we could not identify DNMT3 enzymes in any green algae in MMETSP, although it is present in red algae as it is found in the genomes of *Cyanidioschyzon merolae* and *Galdieria sul-phuraria* (Fig. 2, Supplementary Data 2). We also report several additional DNMT3 transcripts in dinoflagellates, as previously described[26] (Fig. 2, Supplementary Data 2). Upon alignment, dinoflagellate DNMT3 enzymes (including former annotated enzymes[26]) and *Goniomonas pacifica* DNMT3s are closely related

to those from red algae but diverge from other DNMT3s, while diatoms display their own DNMT3 family (Supplementary Fig. 1). This suggests that the DNMT3 family was iteratively lost and acquired several times during microalgal evolution. As observed in *P. tricornutum*, DNMT3 enzymes found in micro-algae, all lack chromatin associated domains (Fig. 1b, Supple-mentary Data 4). This contrasts with mammalian DNMT3s[42] that interact with histone post-translational modifications.

DNMT6 enzymes were found among the most widespread DNMTs in microeukaryotes. We found a DNMT6 transcript in the MMETSP transcriptomes of three *Tetraselmis* green algae and seven dinoflagellates (Fig. 2, Supplementary Data 2). In addition, DNMT6 is distributed extensively in stramenopiles, including *Dictyochophyceae*, *Crysophyceae*, and *Pelagophyceae* (Fig. 2, Supplementary Data 2). In diatoms, DNMT6 is very abundant (Supplementary Data 3). DNMT6 is also present in the non-photosynthetic labyrinthulomycetes *Aplanochytrium stocchinoi* and probably in *Aplanochytrium keurgelense* (Fig. 2, Supplemen-tary Data 2). In addition, our data strongly support the presence of DNMT6 orthologues in the major *Chromalveolata* lineage of *Rhizaria* (Fig. 2, Supplementary Data 2), as suggested in previous reports[26]. DNMT6 enzymes are mostly homogeneous and do not contain chromatin associated signatures, as in *P. tricornutum* DNMT6 and DNMT3 (Fig. 1b, Supplementary Data 4). Finally, monophyletic relationships within the DNMT6 family and

between microeukaryotes could not be resolved (Supplementary Fig. 1).

**Single base resolution of DNA methylation in *P. tricornutum* DNMT5:KO cell lines**. The pennate diatom *P. tricornutum* is the model diatom species that we and others use to study the epigenomic landscape in diatoms, shedding light into the conservation and divergence of DNA methylation patterns in early diverging eukaryotes[27,43]. The *P. tricornutum* genome encodes DNMT3 (Phatr3_J47136), DNMT6 (Phatr3_J47357) and DNMT5a (Phatr3_EG02369) orthologues in single copies but lacks the DNMT4 and DNMT5b orthologues found in other diatoms (Supplementary Data 3). We asked whether any of these DNMTs have DNA methylation function(s) in vivo. Using a CRISPR/Cas9-mediated knockout approach, we screened *P. tricornutum* for DNMT loss of function mutants (see Methods section). In this work, we report five independent mutants with homozygous out-of-frame deletions generating premature STOP codons in the coding sequence of DNMT5a named 'M23', 'M25', '7C6', '7C7' and 'M26' DNMT5:KOs. Two mutants, M23 and M25 with frameshift in the coding sequence of DNMT5 were retained for further investigation (Supplementary Fig. 3). Those mutants are also homozygous for other polymorphisms in the region (Supplementary Fig. 3) that could suggest gene conversion after Cas9 double strand breaks in *P. tricornutum*[44]. No DNMT3 or DNMT6 mutations could be generated using the CRISPR/Cas9 editing strategy.

Using sets of primer pairs targeting the DNMT domain as well as the DEADX helicase-SNF2 like domain of DNMT5 transcripts, we detected by RT-qPCR a 4- to 5-fold loss in mRNA levels in both M23 and M25 cell lines (Supplementary Fig. 4a, Supplementary Data 5). 5mC dot blot screening revealed that all DNMT5:KOs had a 4–5 fold loss of DNA methylation compared to the Pt1 8.6 reference ('wild-type') (Supplementary Figs. 4b, 5), consistent with the putative role of DNMT5 in maintaining DNA methylation patterns in diatoms.

To generate a quantitative single base resolution of DNA methylation loss in DNMT5:KOs, we performed whole genome bisulfite sequencing in M23, M25 (considered as two biological replicates) and the reference, Pt1 8.6 line. We filtered cytosines by coverage depth considering a 5X coverage in all cell lines as a threshold and computed CG methylation levels in TEs and genes. We found that CG methylation is severely impaired in M23 and M25 compared to Pt1 8.6 cell line (Fig. 3a, b, Supplementary Data 6). This is particularly observed within TEs that are the targets of DNA methylation in *P. tricornutum* (Fig. 3a, b). To get a quantitative view of the loss of DNA methylation in DNMT5:KOs, we defined differentially methylated regions (DMRs). We computed DMRs between DNMT5:KOs and WT lines using the bins built-in DMRcaller[45] tools considering 100 bp bins with a minimal difference of $+/- 20\%$ DNA methylation at CGs (5X coverage) in mutants compared to the Pt1 8.6 line. Those thresholds were used based on the minimum coverage per cytosine and the methylation characteristics in our sequencing data (Supplementary Fig. 6a, b). We identified 1715 and 1720 CG DMRs in M23 and M25, respectively (Supplementary Data 7 and Supplementary Data 8), of which 96% are shared between both mutants and show a consistent loss of DNA methylation upon knockout of DNMT5a (Fig. 3c), referred in this study as common hypoDMRs. CG common hypoDMRs cover ~0.8% of the *P. tricornutum* genome. According to the distribution of DNA methylation in the reference strain, we found that 14.90% ($n = 454$) of annotated TEs are found within common hypoDMRs (Fig. 3d, Supplementary Data 9). In order to take into account the possible methylation loss occurring in regulatory

regions, gene and TE coordinates were extended by 500 bp and 1 kb, respectively, upstream and downstream of their start and end sites, considering that intergenic length in *P. tricornutum* varies between 1 kb and 1.5 kb[27]. As a result, respectively, 7.76% and 12.23% of TEs are found within 500 bp and 1 kb of common hypoDMR coordinates (Fig. 3d, Supplementary Data 9). Consistent with their low level of CG DNA methylation observed in both cell lines, we found a comparatively low overlap of common hypoDMRs with genes or their regulatory regions (Fig. 3d, Supplementary Data 9). We then asked whether these common hypoDMRs associate with known regions marked by histone post-translational modifications. Genomic coordinates of common hypoDMRs overlapped with previously mapped histone post-translational modification peaks[43]. The number of common hypoDMRs overlapping with each combination of histone marks is shown in Fig. 3e (Supplementary Data 10). Interestingly, we found that between 80 and 90% of these common hypoDMRs (set size >1500, Fig. 3e) overlap with known regions marked by H3K27me3, H3K9me3, or H3K9me2 defined in the reference Pt1 8.6 line[43]. In addition, 963 (53%) of the common hypoDMRs are found within regions co-marked by all three repressive histone marks (Fig. 3e). This is consistent with the observation that highly methylated regions described by restriction methylation-sensitive sequencing (Mcrbc-Chip) also associate with such histone marks[27]. Our data are consistent with a global loss of DNA methylation in DNMT5:KOs at TE-rich DNA methylated-H3K27me3, H3K9me2 and H3K9me3 marked regions in the *P. tricornutum* genome.

**Gene and TE expression in the absence of DNMT5a in *P. tricornutum***. The control of TEs by the DNA methyltransferase family is a key unifying feature within eukaryotes[2]. Therefore, we monitored the transcriptional effect of DNMT5a loss on genes in M23 and M25 backgrounds using whole RNA high throughput sequencing (see Methods section). Given the high level of DNA methylation observed at TEs compared to genes, we asked whether our RNAseq data captured any TE overexpression that could be linked to hypoDMRs. We thus analyzed TE-gene transcripts that correspond to the expression of TE open reading frames (i.e., encoding reverse transcriptase and integrases) but also genes with TE insertions (Fig. 4a), domesticated TEs and mis-annotated TE loci[27,46]. To identify the most significant changes in mRNA levels, we focused our analysis on genes and TE-genes showing a significant 2-fold induction or reduction of expression in mutants compared to the reference line ($|LFC| > 1$ and an FDR < 0.01, Supplementary Data 11). In M23 and M25, respectively, a total of 1732 and 806 genes and TE-genes are overexpressed while downregulation was observed for 1152 and 248 genes and TE-genes (Fig. 4b). Stable expression ($-1 < LFC < 1$ and FDR < 0.01) is observed for 943 genes and TE-genes in M23 and 216 genes and TE-genes in M25. We found that 557 genes are overexpressed in both cell lines (M23 ∩ M25). A total of 225 genes are overexpressed in M25 only (M25-spe) and 1126 are overexpressed in M23 only (M23-spe). Significantly upregulated genes in both mutants show consistent overexpression levels (Fig. 4c).

In line with the hypothesis that TEs and not genes are directly regulated by DNA methylation in *P. tricornutum*, we found that 338 TE-genes are upregulated in both mutants (Fig. 4d), which correspond to 56% of overexpressed TE-genes. Gene ontology (GO) analysis showed that the upregulated TE-genes are enriched in DNA integration biological function indicating that they mainly correspond to bona fide TE annotations (Fig. 4d, Supplementary Data 12). In contrast, only 219 (16%) of protein coding genes are overexpressed in both mutants. They show clear enrichment for GOs associated with protein folding as well as

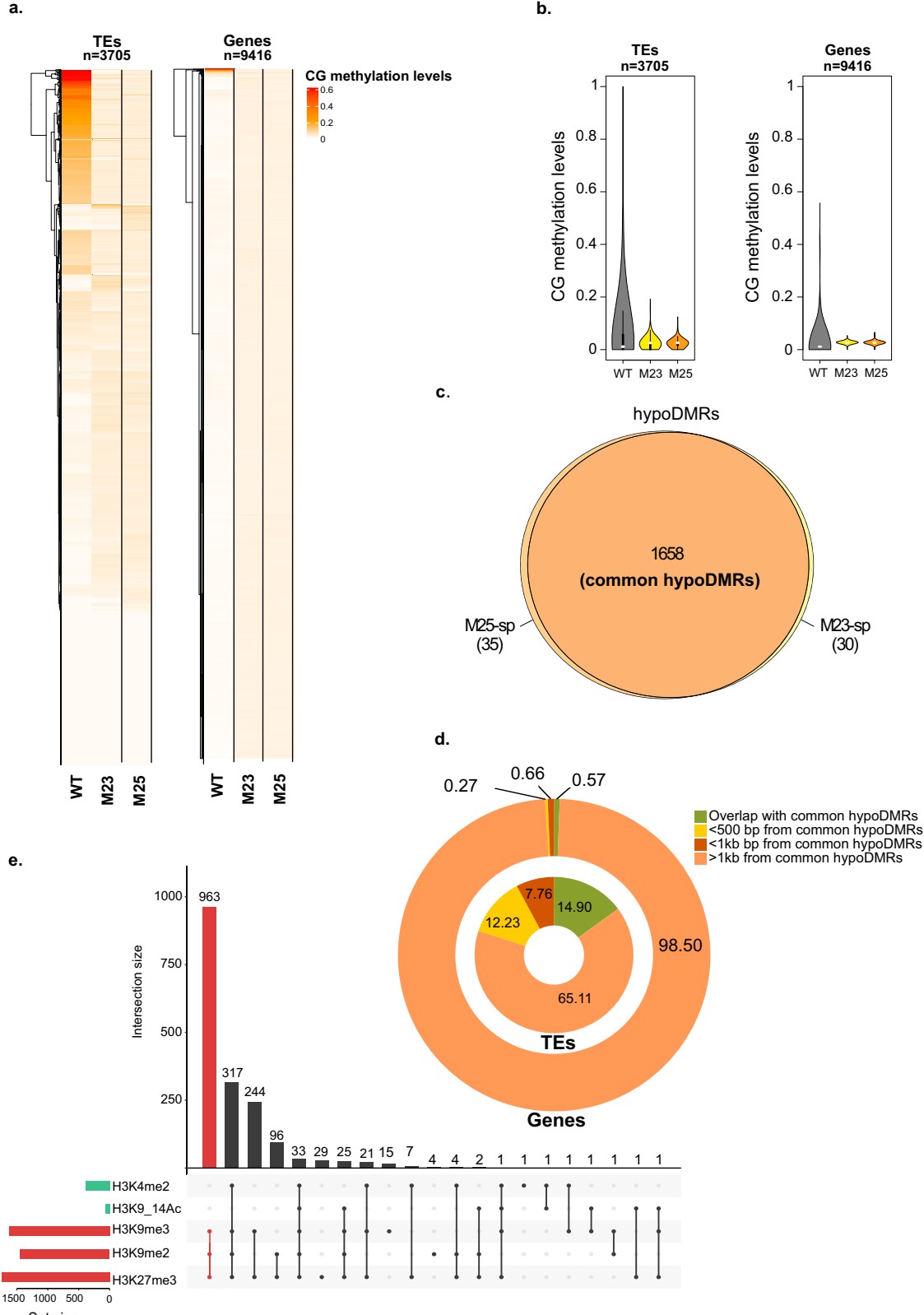

**Fig. 3 CG methylation profiles in DNMT5:KO cell lines. a** Heatmap of CG methylation levels in Pt1 8.6 reference (WT), M23 and M25 for TEs (left panel) and genes (right panel). **b** Violin plot showing the methylation levels in all CGs found in TEs and genes in Pt1 8.6 and M25, M23. Box plots: the bottom of the box displays the lower quartile, the upper part, the upper quartile and the white dots show the median. **c** Venn diagram displaying the number of hypoDMRs identified in M23 (M23-spe) (yellow) and M25 (M25-spe) (orange). **d** Percentages of overlap between common hypoDMRs, genes and TEs. **e** Association between common hypoDMRs and regions targeted by histone post-translational modifications representative of the epigenetic landscape of *P. tricornutum*. The number of overlapping common hypoDMRs is shown for each histone modification and each combination of histone modifications.

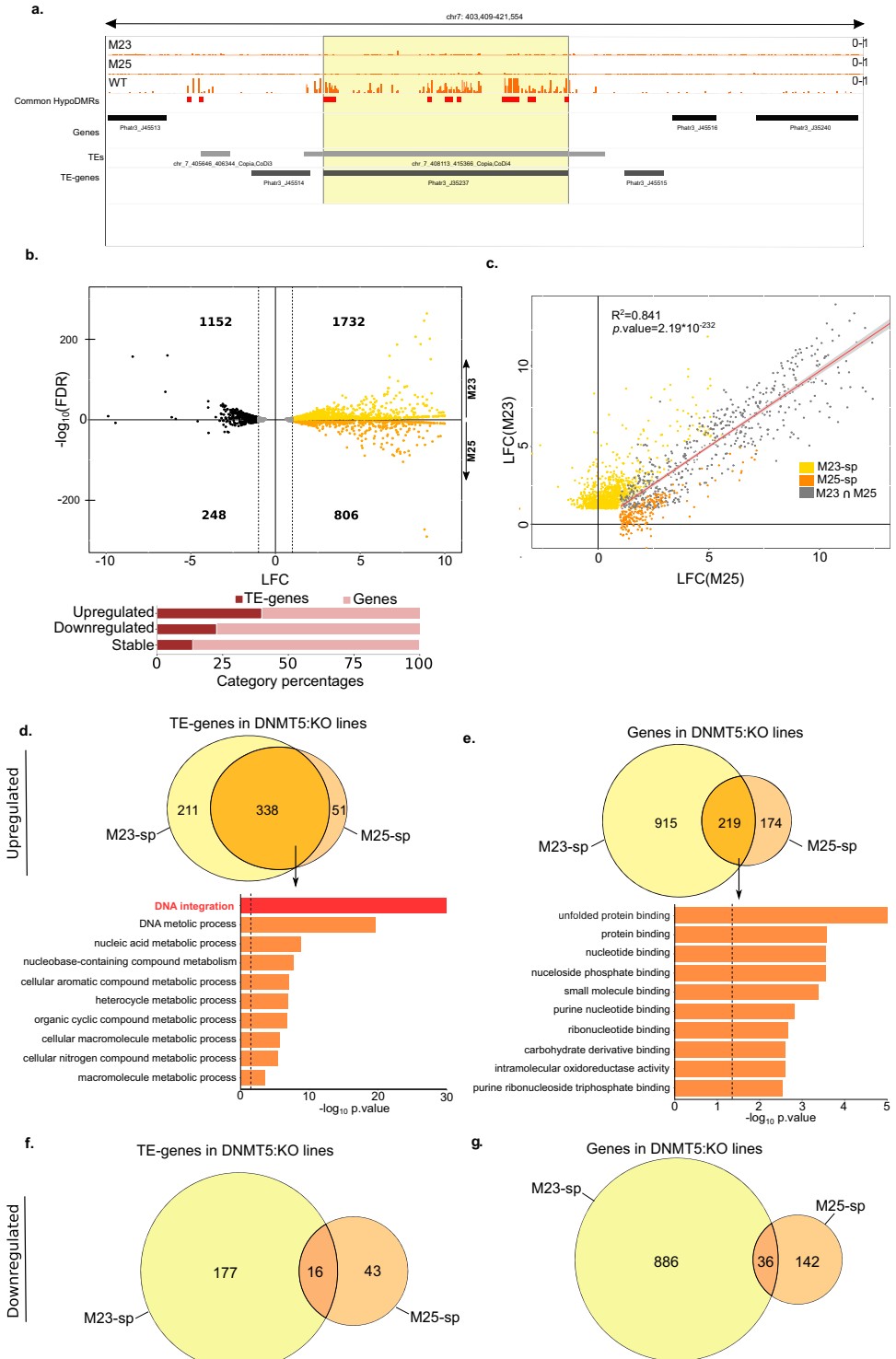

nucleotide phosphate metabolism and nucleotide binding activity (Fig. 4e, Supplementary Data 12). This is typified by the overexpression of chaperone DnaJ domain-containing proteins and Hsp90-like proteins (Supplementary Data 13). The down-regulation of genes was not consistent between M23 and M25 as only 35 genes and 16 TE-genes are downregulated in both cell lines (Fig. 4f, g, Supplementary Data 14). Expression levels of 12 genes was confirmed by qPCR in the M23 cell line, including DnaJ and HSP90-like protein coding genes mentioned previously (Supplementary Fig. 7a, b, Supplementary Data 15). Of note DNMT5a (Phatr3_EG02369) is among the downregulated genes

in both mutants (Supplementary Data 14), consistent with qPCR results.

**Relationship between CG methylation and expression of TE-genes in *P. tricornutum*.** The observed overexpression of TEs in DNMT5:KOs could be directly due to the loss of DNA methylation. To test this, we first determined DNA methylation levels in the 600 upregulated TE-genes in the DNMT5:KO lines (Fig. 5a, Supplementary Data 16). For each TE, we also computed the mean-centered normalized LFC (z-score) for each of the M23 and M25 lines (Fig. 5a, Supplementary Data 16). We found that the

**Fig. 4 Dynamics of gene and TE-gene expression in DNMT5:KO cell lines. a** Snapshot of an example TE-gene CG methylation profile. **b** Differential expression in DNMT5:KOs (M23 and M25 are represented in the upper and lower parts of the volcano plot, respectively) compared with Pt1 8.6 reference (WT). The upper panel shows a volcano plot that displays the distribution of the fold changes (LFC) in the $X$-axis and adjusted $p$.values (-log10FDR) in the $Y$-axis. The number of genes up and downregulated in each mutant are indicated. The stable genes (1 < LFC > −1 and FDR < 0.01) are shown in gray. The lower panel shows a barplot that displays the proportion of genes and TE-genes in each expression category (downregulated, stable and upregulated). **c** Scatter plot comparing fold changes of M23 and M25 upregulated genes. Yellow and orange dots represent specific significantly upregulated genes in M23, M25, respectively (LFC > 1 and FDR < 0.01, M23-sp, M25-sp, respectively). Gray dots represent significantly upregulated genes in both mutants (LFC > 1 and FDR < 0.01, M23 ∩ M25). The solid line represents the linear fit and the gray shading represents 95% confidence interval for the significantly upregulated genes in both mutants. **d**, **e** The upper panel represent Venn diagrams showing the numbers of specific (M23-sp, M25-sp) and common (M23 ∩ M25) upregulated TE-genes and genes, respectively, in each mutant compared to the Pt1 8.6 reference (WT). The lower panel shows the top 10 enriched canonical pathways of upregulated TE-genes and genes, respectively, sorted by p.value in both mutants (M23 ∩ M25) as identified by topGO analysis. The dashed lines show the p.value of 0.05. **f** Venn diagram displaying downregulated TE-genes (LFC < −1 and FDR < 0.05) in M23 (M23-spe) (yellow) and M25 (M25-spe) (orange). **g** as for **f** for downregulated genes.

TE-genes with the highest LFC (z-score > 2) in the mutants are associated with higher DNA methylation levels in the reference strain. This is the case for each mutant independently, indicating that TEs with the highest upregulation in the DNMT5:KO lines are direct targets of DNA methylation in the reference strain.

We then assessed the relationship between upregulated TE-genes and the common hypoDMRs and found that 62% of upregulated TE-genes are found within these DMRs (Fig. 5b). Importantly, this was the case only for TE-genes with over-expression in both cell lines (M23 ∩ M25) and not for M23-spe and M25-spe upregulated TE-genes (Fig. 5b). This also means that 40% of upregulated TE-genes cannot be explained by the loss of DNA methylation alone. Similarly, downregulation and stable expression are not associated with common hypoDMRs (Fig. 5b). This shows that TE-genes with consistent upregulation are specifically due to the loss of DNA methylation while other TE-gene misregulation is due to cell line specific DNA methylation-independent regulation. Among the 128 upregulated TE-genes in both mutants that are not direct targets of DNA methylation, we found a common hypoDMR in the regulatory region of 42 (in M23) and 15 TE-genes (in M25), respectively, indicating that DNA methylation loss at these regions was also responsible for their upregulation (Fig. 5c).

Next, we assessed TE families as annotated previously[46] (Fig. 5d). We find that overexpressed TE-genes are mostly represented by Copia-like in diatoms (CoDi) retrotransposons of the CoDi1, CoDi2, CoDi4, and CoDi5 families with a minority of DNA transposons as the PiggyBack family (Fig. 5d). We noticed that the TE families are found in similar proportions among TEs that overlap the common hypoDMRs and those that do not. However, when we compared TE lengths, TEs that are upregulated and overlap with common hypoDMRs are longer than upregulated TEs that are not overlapping with hypoDMRs (Fig. 5e, Supplementary Data 17). This suggests that younger TEs tend to be direct targets of DNA methylation compared to evolutionary older TEs family members. Subsequently, loss of DNA methylation causes upregulation of mainly younger TEs. Filloramo et al.[47] recently described 85 long-LTR-copia-like (LTR-copia) TEs based on reannotation of the *P. tricornutum* genome by Oxford Nanopore Technologies long-read sequencing. Such TEs are considered as potentially still active[47]. They are represented by Copia-like in diatoms (CoDi) of the CoDi5, CoDi4 and CoDi2 families[47] that corresponds to the TE families found overexpressed in our datasets (Fig. 5d). Accordingly, we found that 75/85 of LTR-copia are targets of DNA methylation and are associated with common hypoDMRs (Supplementary Data 18). In addition, by overlapping TE-genes and genomic locations of LTR-copia, we found that 61/75 of LTR-copia are overexpressed in both mutants (Supplementary Data 18). Of note, our RNAseq data thus also support the presence of these TEs in the reference

Pt1 8.6 cell line as potentially still active elements. An example of upregulation at LTR-copia is shown in Fig. 5f. Additional shorter TEs with overexpression also belong to CoDi5, CoDi4 and CoDi2 TE categories suggesting that an active expression might still remain. Altogether, this strongly suggests that DNA methylation is involved in the repression of young TEs in the *P. tricornutum* genome.

## Discussion
Studies on the evolutionary history of DNMTs have established that the DNA methylation machinery diverged among eukaryotes along with their respective DNA methylation patterns[2,11]. However, the diversity of DNMTs found in SAR lineages is under-explored due to the lack of representative sequences. Based on MMETSP transcriptomes, we set out to explore the diversity and phylogeny of DNMTs in early diverging eukaryotes. Besides the absence of genomic sequences, the MMETSP database only encompasses expressed transcripts from cultured organisms and is thus deprived of lowly expressed genes and condition-specific expressed genes. Absence of a given gene family within a species should therefore be interpreted accordingly. When our analysis found multiple distinct transcripts sharing the same DNMT subfamily, as in diatoms, we used the most probable open reading frame translation of the transcripts using eDAF curation to produce our phylogenetic tree. However, without genomic annotations, we cannot rule out that such transcripts result from alternative transcription originating from a single gene or multi-copy gene families. Our data are best interpreted when multiple transcripts and annotated genes, whenever possible, are available rather than at the species-specific level.

We nonetheless confirm that stramenopiles and dinoflagellates encode a divergent set of DNMT proteins including DNMT3 and DNMT6, which have no chromatin associated domains. In addition, our study independently reports the same DNMT6 enzymes found in the raphidophyceae, *Bigelowella natans* and *Aplanochytrium stochhinoi* by earlier work although not specified by the authors[26]. As reported in trypanosomes[10], we suggest that DNMT6 likely emerged prior to the *Chromalveolata* radiation. In trypanosomes, its presence in several lineages does not predict DNA methylation per se and must be further investigated[48].

The DNMT5 enzymes are also very well represented both at the genomic and transcriptomic levels, even outside the SARs, and are thus likely ancestral to eukaryotes. We show here that the DNMT domains among the different DNMT5s are conserved but show a divergence compared to other DNMTs, thus supporting a common evolutionary origin for all DNMT5 enzymes. DNMT5b enzymes could be multifunctional enzymes as suggested by the presence of N-terminal HAND domains found in chromatin remodelers[49], by TUDOR domains found in histone modifying enzymes, histone post-translational modification readers[50] as well as small RNA

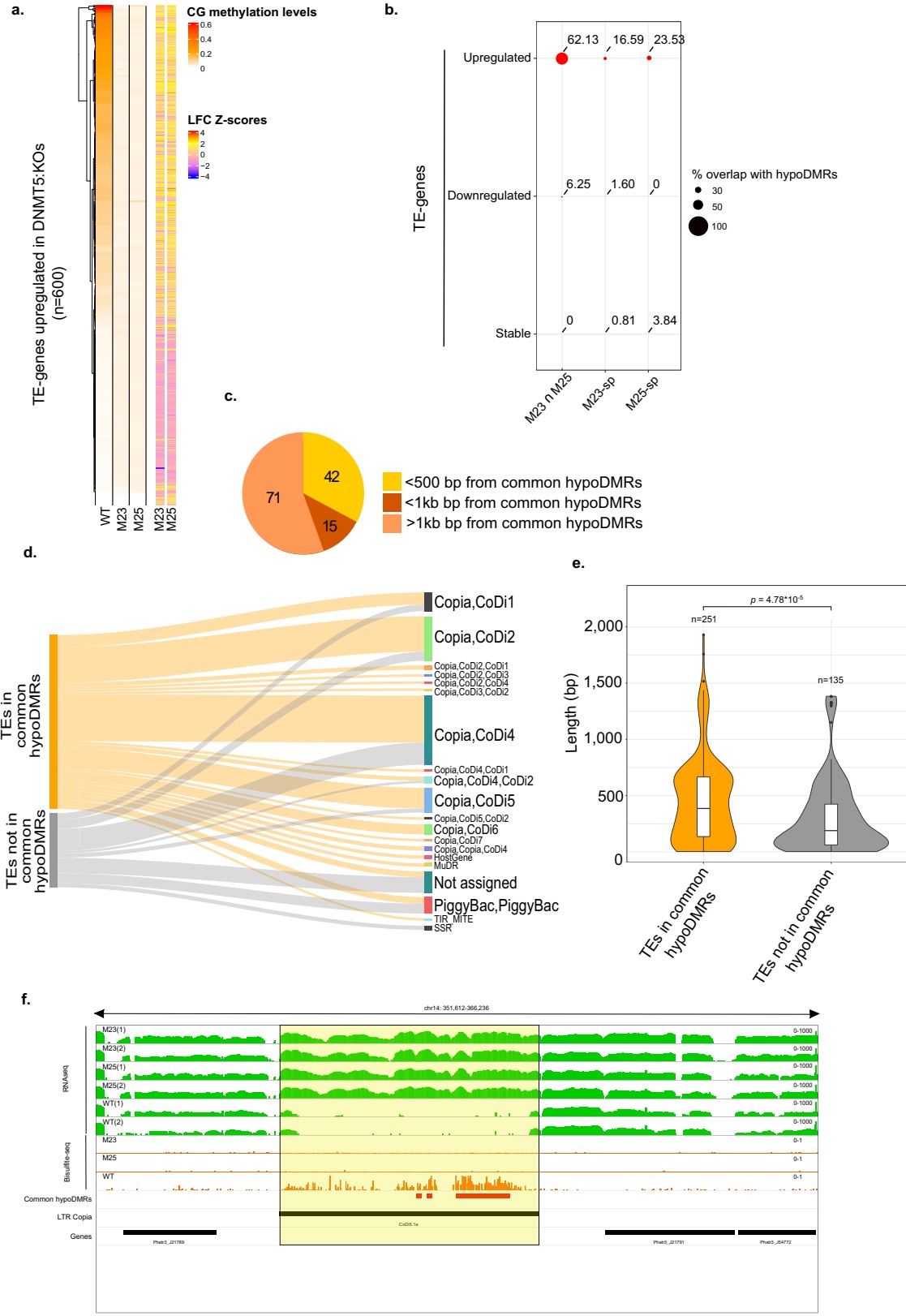

interacting proteins[51,52]. They also contain an SNF2 ATPase domain[11] which plays a chaperone-like enzyme-remodeling role important for DNA methylation and its targeting to specific sites[15,53]. DNMT5c enzymes are likely very divergent DNMT5 enzymes that lack ATPase SNF domains. The diversity of DNMT5 domains is likely inherent to its functioning and interaction with

other epigenetic processes such as histone modifications and non-coding RNA. In mammalian cells, TUDOR domain containing UHRF1 is known to target DNMT1, the functional homologue of DNMT5, onto newly synthesized DNA substrates during semi conservative DNA replication[54]. Furthermore, TUDOR domain of UHRF1 was reported to play an important role in the recognition

**Fig. 5 Interplay between CG methylation and TE-gene expression. a** Heatmap of CG methylation levels in Pt1 8.6 reference (WT), M23 and M25 (left panel) and LFC normalized levels (Z-scores) (right panel) of the 600 upregulated TE-genes in M23 and M25 compared to Pt1 8.6 reference (WT). **b** Percentages of overlap between common hypoDMRs and upregulated (red), downregulated (blue) and stable TE-genes (gray) in M23 only (M23-sp), M25 only (M25-sp) and both mutants (M23 ∩ M25). **c** Distribution of common (M23 ∩ M25) upregulated TE-genes that overlap with TE-genes regulatory regions. **d** Mapping of TEs covered in TE-genes that overlap or not with common hypoDMRs (queries in the left) onto annotated TEs on Phatr3. Bar sizes are proportional to the number of TEs in the queries that are assigned to each TE category. **e** Violin plot comparing the length (bp) of TEs covered in TE-genes that overlap or not with common hypoDMRs, Wilcoxon test. The white boxplots show summary statistics, the bottom of the box displays the lower quartile, the upper part, the upper quartile and the middle line the median. **f** IGV snapshot of expression levels in both replicates of WT and DNMT5:KOs (M23 and M25) (green tracks) and CG methylation levels in the WT and DNMT5:KOs (M23 and M25) (orange tracks) of an example LTR-copia (highlighted in yellow). The common hypoDMRs and genes are also shown in the red and black tracks, respectively.

of histone H3K9 methylation[55,56]. While UHRF1, DNMT1, and ATPase protein containing domains are separate in animals, they form an unusual multifunctional domain protein in DNMT5 in photosynthetic microeukaryotes. This domain architecture might be due to the compact genomes of the green microalgae *Micromonas pusilla*, *Ostreoccocus tauri* and *Bathycoccus prasinos*[11].

In our phylogenetic study, the RID/DMTA and diatom DNMT4 enzymes are related, as shown previously by Huff and Zilberman[11] and Punger and Li[10]. In our case, because the analysis covers a large evolutionary distance, phylogenetic relationships between DNMT families should be interpreted accordingly. Therefore, we cannot rule out the possibility that diatoms and RID families are paraphyletic. The function of DNMT4 or DNMT4-type enzymes in diatoms is unknown. Among the four diatoms with a known methylation pattern on TEs, two are lacking DNMT4s (including *P. tricornutum* presented in this study). The presence of chromodomains known to bind histone post-translational modifications as in CMT enzymes[57] nonetheless suggests that diatom DNMT4 might be functional as either a *de novo* or a maintenance enzyme. The lack of chromatin-associated domains in DNMT3, DNMT6, and other DNMT4 proteins suggest that the link, if any, between DNA methylation and histone modifications is more indirect than observed in plants and mammals and might require the activity of accessory proteins like UHRF1-type[54] or DNMT3-like[58] enzymes that should be further investigated.

Examining the role of DNMT5a in the pennate diatom *P. tricornutum*, we found that it is an orthologue of the single DNMT5a protein from *Cryptococcus neoformans*, which is involved in the maintenance of DNA methylation[11,15]. In that regard, our study demonstrates that the loss of DNMT5a was sufficient alone to generate a global loss of CG methylation in *P. tricornutum* similar to *Cryptococcus neoformans*[11]. We further confirm that TEs are major targets of DNA methylation in diatoms. In addition, DMR analysis identified regions essentially composed of TEs that show extensive methylation in the reference strain. HypoDMRs overlap with regions marked by H3K27me3 but also H3K9me3 which suggest that histone post-translational modifications and DNA methylation cooperate to maintain TE repression. Genes appear not to be the primary targets of DNA methylation. Only 51 out of all 9416 annotated genes are found within DMRs. Among them, 19 were upregulated in both KO mutants. TE methylation is observed in other diatoms such as *F. cylindrus*[11] and *T. pseudonana*[11] where the targeted TEs have low expression[11]. However, those species encode a different set of DNMTs compared to *P. tricornutum*. *T. pseudonana* appears to lack DNMT5a and has a partial DNMT6 protein while *F. cylindrus* encodes all but DNMT3 (Supplementary Data 3). It is possible that DNMTs show partial functional redundancy in diatoms. In that regard, the DNMT5:KO cell lines presented in this study could be used as a heterologous expression system to decipher the role of other DNMTs in diatoms.

Compared to DNA methylation loss that is observed in different DNMT5:KO cell lines (Supplementary Figs. 4, 5), gene expression was more inconsistent between cell lines, including when assessed by qPCR validation. We thus make the hypothesis that gene expression is mainly cell line specific in DNMT5:KO cell lines. This divergence in gene expression could be linked to the random insertions of plasmids generated by biolistic transformation. Alternatively, *de novo* and likely random TE insertions upon DNA methylation loss could generate gene expression divergence between cell lines over time.

In our study, we found that 15% of TE-genes are upregulated in the DNMT5:KO cell lines, less than observed in *Arabidopsis thaliana* where the loss of DDM1 (involved in the maintenance of DNA methylation) caused the expression of about 40% of all TE-genes[59]. However, in *P. tricornutum* we found that overexpression and methylation levels are particularly relevant for TEs that have been identified as full length potentially still active LTR-copia elements. Interestingly, in *Arabidopsis thaliana*, the most mobile TEs between different accessions are regulated by the MET2a protein, likely involved in DNA methylation and repression[60]. In addition, such TEs expansion associates with null or loss of function alleles of MET2a[60]. When comparing *P. tricornutum* and *T. pseudonana* genomes, the CoDi2 and CoDi4 families are the main contributors of retrotransposon expansion in *P. tricornutum*[61] although CoDi2 is only found in *P. tricornutum*. We found such TEs to be overexpressed in response to DNA methylation loss. Therefore, DNA methylation seems to be a genome integrity keeper in *P. tricornutum*. Other smaller TEs in the form of TE-genes are also upregulated and may retain some activity in *P. tricornutum*. Upregulation was also observed for TEs that were not targets of DNA methylation in the reference strain but for which a subset was nonetheless found within a 1 kb distance from hypoDMRs, suggesting that initial repression is likely linked to DNA methylation spreading or proximity, which was reported in a previous work[27].

Highly repetitive TE families are removed in our analysis since only uniquely mapped reads were aligned. This is true for both transcriptomic and bisulfite sequencing data. In addition, our transcriptomic analysis is only a snapshot of all TEs overexpressed at a given time in *P. tricornutum* cell populations. The loss of DNA methylation could trigger more misregulation of TEs in stress culture conditions, as previously reported upon nitrogen depletion[27] and exposure to the toxic reactive aldehyde[61]. DNMT5 mutant cell lines are viable in standard culture conditions used for *P. tricornutum* suggesting that co-occurring repressive histone marks reported in previous studies might be compensating the loss of DNA methylation[35,43]. Additionally, this suggests that in optimal conditions, loss of DNA methylation is not associated with drastic biological effects, supporting the lack of a phenotypic response which is otherwise seen in stress conditions, typically slow growth, smaller cell size and an atypical morphology. Our study provides direct understanding of DNA methylation regulation and its role in diatoms offering a robust

foundation for future studies in eukaryotes to further comprehend DNA methylation function and evolution.

## Methods

**HMMER and Reciprocal BLAST best Hit analysis**. We performed an extensive scan of the MMETSP database, enriched with 7 diatom transcriptomes and genomes from the top 20 most abundant diatoms found in *Tara* Oceans database[62], using HMMER-search with the model PF00145 to fetch any DNMT-like, including partial transcripts, sequence within microeukaryotes. We ran HMMER in a non-stringent fashion to not miss positives DNMT sequences. We used eDAF approach to filter the expected high number of false positives. It is worth noting that we initially use HMMER for screening instead of the built-in module of eDAF due to the time complexity of the latter for extensive searches (tens to hundreds of times slower than HMMER). Reciprocal BLAST best hit analysis was performed as previously described[63]. Briefly, the DNMT3 (Phatr3_J47136), DNMT4 (*Thaps3_11011*), DNMT5 (*Phatr3_EG02369*) and DNMT6 (Phatr3_J47357) orthologues found in *P. tricornutum* or *T. pseudonana* (for DNMT4) were blasted against a phylogenetically optimized database that include MMETSP transcriptomes. Putative DNMT sequence hits giving back the corresponding enzyme (DNMT3, DNMT4, DNMT5, or DNMT6) at the threshold of e-value of $1 \times 10^{-5}$ in the corresponding diatom were retained. Candidate enzymes were then analyzed using eDAF.

**eDAF-guided domain architecture analysis**. enhanced Domain Architecture Framework (eDAF) is a four-module computational tool for gene prediction, gene ontology, and functional domain predictions[35]. As previously described for Polycomb and Trithorax enzymes[35], candidate DNMTs identified by RBH and HMMER-search were analyzed using the DAMA-CLADE built-in functional domain architecture to predict the occurrences of DNMT domains starting from protein sequences. The domain architecture of representative enzymes used in this study can be found in Supplementary Data 4.

**Phylogenetic tree analysis**. The DNMT domain of candidate enzymes were aligned using ClustalΩ[64] (HHalign algorithm). The alignment was manually curated and trimmed using trimAL (removing >25% gap column) to align corresponding DNMT motifs in all gene families. A convergent phylogenetic tree was then generated using the online CIPRES Science gateway program[65] using MrBAYES built-in algorithm. Default parameters were used with the following specifications for calculation of the posterior probability of partition: sumt.burninfraction = 0.5, sump.burningfraction = 0.5, 10000000 generations, sampling each 100. We also used two different models: Estimating the Fixed Rate and GTR.

**Cell cultures**. Axenic *P. tricornutum* CCMP2561 clone Pt1 8.6 cultures were obtained from the culture collection of the Provasoli-Guillard National Center for Culture of Marine Phytoplankton (Bigelow Laboratory for Ocean Sciences, USA.). Cultures were grown in autoclaved and filtered (0.22 μM) Enhanced Sea Artificial Water (ESAW - https://biocyclopedia.com/index/algae/algal_culturing/esaw_medium_composition.php) medium supplemented with f/2 nutrients and vitamins without silica under constant shaking (100 rpm). Cultures were maintained in flasks at exponential state in a controlled growth chamber at 19 °C under cool white fluorescent lights at 100 μE m − 2 s − 1 with a 12 h photoperiod. For RNA sequencing and bisulfite experiments, WT and DNMT5 mutant cultures were seeded in duplicate at 10.000 cells/ml and grown side by side in 250 ml flasks until early-exponential at 1.000.000 cells/ml. Culture growth was measured using a hematocytometer (Fisher Scientific, Pittsburgh, PA, USA). Pellets were collected by centrifugation (4000 rpm) washed twice with marine PBS (http://cshprotocols.cshlp.org/content/2006/1/pdb.rec8303) and flash frozen in liquid nitrogen. Cell pellets were kept at −80 °C until use. For bisulfite sequencing, technical duplicates were pooled to get sufficient materials.

**CRISPR/Cas9 mediated gene disruption**. The CRISPR/Cas9 knockouts were performed as previously described[44]. Our strategy consisted in the generation of short deletions and insertions to disrupt the open reading frame of putative DNMTs of *P. tricornutum*. We introduced by biolistic the guide RNAs independently of the Cas9 and ShBle plasmids, conferring resistance to Phleomycin, into the reference strain Pt1 8.6 (referred hereafter as 'reference line' or 'wild-type'-WT). Briefly, specific target guide RNAs were designed in the first exon of Phatr3_EG02369 (DNMT5), Phatr3_J47357 (DNMT6) and Phatr3_J36137 (DNMT3) using the PHYTO/CRISPR-EX[66] software and cloned into the pU6::AOX-sgRNA plasmid by PCR amplification. For PCR amplification, plasmid sequences were added in 3' of the guide RNA sequence (minus –NGG), which are used in a PCR reaction with the template pU6::AOX-sgRNA. Forward primer – sgRNA seq + GTTTTAGAGCTAGAAATAGC. Reverse primer - sequence to add in 3' reverse sgRNA seq + CGACTTTGAAGGTGTTTTTG. This will amplify a new pU6::AOX-(your_sgRNA). The PCR product is digested by the enzyme DPN1 (NEB) in order to remove the template plasmid and cloned in TOPO10 *E. coli*. The sgRNA plasmid, the pDEST-hCas9-HA and the ShBLE Phleomycin resistance gene cloned into the plasmid pPHAT-eGFP were co-transformed by biolistic in the Pt1

8.6 'Wild Type' strain as described in[44]. We also generated a cell line that was transformed with pPHAT-eGFP and pDEST-hCas9-HA but no guide RNAs. This line is referred as the Cas9:Mock line.

CRISPR/Cas9—sgRNA transformants were selected by phleomycin resistance (carried by the plasmid pPHAT-eGFP). 48 hours post-transformation, diatoms were replated and grown on phleomycin 100 ug/ml 50% Enhanced Artificial Sea Water[67] plates until single colonies appeared (2–3 weeks). Transformants were isolated from plates and sanger sequenced after PCR using HGS Diamond Taq® as per manufacturer instruction and the primers DNMT5locus_R and DNMT5locus_F (Supplementary Data 19). These primers allow the amplification and sequencing of both alleles of DNMT5. Sequencing using the primer DNMT5locus_R shows that mutants are homozygous for the deletions. In addition, all mutants show loss of heterozygosity (LOH).

**RNA and DNA extraction**. Total RNA extraction was performed by classical TRIZOL/Chloroform isolations and precipitation by isopropanol. Frozen cell pellets were extracted at a time in a series of 3 technical extraction/duplicates and pooled. RNA was DNAse treated using DNAse I (ThermoFisher) as per manufacturer's instructions. DNA extraction was performed using the Invitrogen™ Easy-DNA™ gDNA Purification Kit following 'Protocol #3' instructions provided by the manufacturer. Extracted nucleic acids were measured using QUBIT fluorometer and NANODROP spectrometer. RNA and gDNA Integrity were controlled by electrophoresis on 1% agarose gels.

**RT-qPCR analysis**. qPCR primers were designed using the online PRIMER3 program v0.4.0 defining 110–150 amplicon size and annealing temperature between 58 °C and 62 °C. Primer specificity was checked by BLAST on *P. tricornutum* genome at ENSEMBL (http://protists.ensembl.org/Phaeodactylum_tricornutum/Info/Index). For cDNA synthesis, 1 μg of total RNA was reverse transcribed using the SuperScript™ III First-Strand (Invitrogen) protocol. For quantitative reverse transcription polymerase chain reaction (RT-qPCR) analysis, cDNA was amplified using SYBR Premix ExTaq (Takara, Madison, WI, USA) according to manufacturer's instructions. CT values for genes of interest were generated on a Roche lightcycler® 480 qpcr system. CT values were normalized on housekeeping genes using the deltaCT method. Normalized CT values for amplifications using multiple couples of primers targeting several cDNA regions of the genes of interest were then averaged and used as RNA levels proxies. Primers used in this study are listed in Supplementary Data 19.

**Dot blot**. gDNA samples were boiled at 95 °C for 10 min for denaturation. Samples were immediately placed on ice for 5 min, and 250–500 ng were loaded on regular nitrocellulose membranes. DNA was then autocrosslinked in a UVC 500 crosslinker – 2 times at 1200uj (*100). The membranes were blocked for 1 h in 5% PBST-BSA. Membranes were probed for 1 h at room temperature or overnight at 4 °C with 1:1000 dilution of 5mC antibody (OptimAbtm Anti-5-Methylcytosine – BY-MECY 100). 5mC signals were revealed using 1:5000 dilution of HRP conjugated antirabbit IgG secondary antibody for 1 h at room temperature followed by chemo luminescence. Loading was measured using methylene blue staining.

**RNA and bisulfite sequencing**. RNA libraries were prepared by the FASTERIS Company (https://www.fasteris.com). Total RNA was polyA purified and libraries were prepared for illumina NextSeq sequencing technologies. For RNAseq analysis, two biological replicates per mutant were used (M23 and M25). In addition, two biological replicates of a Pt1 8.6 line was sequenced in the same run as a control (total of 6 samples). Bisulfite libraries and treatments were performed by the FASTERIS Company and DNA was sequenced on an Illumina NextSeq instrument. 150 bp paired-end reads were generated with 30X coverage. A 5mC map was also generated in the reference Pt1 8.6 line as a control.

**RNAseq analysis**. 150 bp paired-end sequenced reads were subjected to quality control with FastQC (https://www.bioinformatics.babraham.ac.uk/projects/fastqc). Then, the reads were aligned on the reference genome of *P. tricornutum* (Phatr3)[46] using STAR (v2.5.3a)[68]. Gene expression levels were quantified using HTseq v0.7.2. Differentially expressed genes were analyzed using DESeq2 v1.19.37[69] with the following generalized linear model: ~mutation. FDR values are corrected p.values using the Benjamini–Hochberg method. Genes are designed significant (DEGs) if the |log2FC| > 1 and the FDR < 0.05. GO enrichments were calculated using the overrepresentation Fisher's exact test described in topGO v2.44.0[70]. For each analysis, appropriate DEGs have been used as input and a GO theme is considered as significant if the $p < 0.05$.

**Bisulfite sequencing analysis**. Bisulfite analysis was performed using Bismark-bowtie 2 (https://www.bioinformatics.babraham.ac.uk/projects/bismark/). We used the default Bowtie2 implementation of Bismark with the specifications that only uniquely mapped reads should be aligned. All alignments were performed with high stringency allowing for only one base mismatch ($n = 1$). We also clearly specified that no discordant pairs of the pair-end reads should be aligned. DNA methylation in the CG, CHG, and CHH contexts was calculated by dividing the

total number of aligned methylated reads by the total number of methylated plus un-methylated reads.

**DMR calling.** Differentially methylated regions were called using the DMRcaller R package v1.22.0[45]. Given the low level of correlation of DNA methylation observed in *P. tricornutum*[11,27] and sequencing coverage in all three cell lines, only cytosines with coverage > =5X in all three lines were kept for further analysis and the bins strategy was favored over other built-in DMRcaller tools. DMRs were defined as 100 bp regions with at least an average 20% loss/gain of DNA methylation in either one of the DNMT5:KOs compared to the reference strain. The 'Score test' method was used to calculate statistical significance and threshold was set at $p < 0.01$. In addition, to distinguish isolated differentially methylated cytosines from regions with significant loss of DNA methylation, an hypoDMR must contain at least methylated 2 CpG in the reference strain.

**Overlap with histone modifications and genomic annotations.** Analysis on BED files were performed using bedtools v2.27.1.[71]. Bedtools intersect with default parameters was used to calculate overlap regions of DMRs with genes and TE-genes. Bedtools window has been used to compute the 500 bp and 1 kb upstream and downstream near regions between DMRs, genes and TE-genes.

Percentage overlaps between DMRs as well as the overlap of gene and TE coordinates with histone modifications and DMRs were calculated using the genomation R package v1.22.0[72] and the 'annotateWithFeature' and 'getMembers' functions. For RNAseq analysis, we analyzed the expression of TE-genes as previously described[46]. To define TE-genes in DMRs we crosschecked overlapping TE-genes annotations with bona fide TEs in DMRs using 'annotatewithFeature' function. UpSet plots were computed using UpSetR v1.4.0[73] Heatmaps were produced using the R package ComplexHeatmap (v2.8.0)[74]. Sankey diagram was produced with the R package highcharter (v0.9.4) (https://jkunst.com/highcharter/authors.html). TEs that mapped to less than three members of a TE family were discarded. All R plots were obtained using R version 4.0.3.

**Statistics and reproducibility.** The statistical tests used in this study are indicated in the respective figure legend or in the Methods section. We used the function lm() from R to fit the linear model between LFC in the M23 and M25 cell lines. The pairwise Wilcoxon rank-sum test has been used to check for significant differences in TEs lengths between two distributions using the pairwise wilcox.test() function in R (Fig. 5e). For multiple-hypotheses testing, *p*.values were adjusted with the Benjamini–Hochberg procedure. No statistical methods were used to predetermine sample size.

**Reporting summary.** Further information on research design is available in the Nature Portfolio Reporting Summary linked to this article.

## Data availability

The raw data have been deposited at Gene Expression Omnibus GEO (https://www.ncbi.nlm.nih.gov/geo/query/acc.cgi?acc=GSE186857). Bisulfite sequencing raw data and bigwig files showing methylation rates (#methylated C/#total number of C) in the context of CHH, CHG and CpG, where H: is A, C or T in the WT, M23 and M25 are under the accession number GSE186855. The raw RNA sequencing data and the TPM counting table are under accession GSE186856. Source data underlying figures are presented in Supplementary Data 6 (Fig. 3b), Supplementary Data 9 (Fig. 3d), Supplementary Data 10 (Fig. 3e), Supplementary Data 11 (Fig. 4b), Supplementary Data 12–13 (Fig. 4d, e), Supplementary Data 17 (Fig. 5e). A list of primers used in the study is presented in Supplementary Data 19. A full uncropped blot of the dot blot shown in Supplementary Fig. 4b is presented in the Supplementary Fig. 5. All the cell lines used in the study, including DNMT5:KOs, are available upon request.

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

## Acknowledgements

We thank Catherine Cantrel form IBENS for media preparation. L.T. acknowledges funds from the CNRS, the region of Pays de la Loire (ConnecTalent EPIALG project) and Epicycle ANR project (ANR-19-CE20- 0028-02). C.B. acknowledges funding from the ERC Advanced Awards Diatomite. F.Y. was supported by a PhD fellowship from the Chinese Scholarship Council (CSC-201906310152).

## Author contributions

A.H. and L.T. conceived the study. A.H., F.R.J.V., O.A.M., and L.T. designed the study. L.T. supervised and coordinated the study. A.H. performed the experiments. C.B. contributed to the discussion of the results. AuGe contributed to the bioinformatic analysis. A.G. grew the mutants and extracted RNA for validation experiments. F.Y. performed QPCR work and gene validation analysis. O.A.M. performed and supervised A.H. for the bioinformatic analysis of RNAseq, gene ontology and bisulfite seq data. A.H. performed the DMR analysis under the supervision of O.A.M. F.R.J.V. and A.H. analyzed HMMER, DAMA/CLADE and eDAF data. All authors analyzed and interpreted the data. A.H. and L.T. wrote the manuscript with input from all authors.

## Competing interests

The authors declare no competing interests.
