## [Peer Review File · Communications Biology]

Reviewers' comments:

Reviewer #1 (Remarks to the Author):

I appreciate the combined bioinformatic and laboratory approach in this study, investigating the mechanisms of DNA methylation in diatoms. I agree with the authors in the potential importance and reach of this work and find their methodology to be appropriate. The manuscript is largely well-written and comprehensible.

I would only suggest a more nuanced interpretation of the results, particularly in regard to the evolution and phylogenetic interpretation. I've laid out some of my concerns below.

Introduction

Line 144: Do the authors consider diatoms an "early diverging eukaryote lineage"? I don't know that I would consider a phylogenetically-derived product of a secondary endosymbiotic event that diverged some 200 mya as "early diverging" for a eukaryote...

Results

Figure S1: Taxon identities would be more useful as branch termini to assess this phylogeny than these sequence codes. Are these branches multiple copies of the marker in species, or different species?

Line 182-184: Explain what you mean by this statement. The topology of the enzyme is unconvincing, or the phylogenetic position of these *E. huxleyi* sequences is suspect because they are not monophyletic?

Line 186-187: "Lineage specific loss" is a tough sell here when the "lineages" are so broadly defined as "centric versus pennate". These states are broad morphological categories which do not necessarily correspond to genetic lineages. The "centric" diatoms, for example, are not monophyletic by DNA sequence data and represent several genetic lineages. In fact, among the taxa labeled "centric" in Table S3, almost 65% are from the same taxonomic order—the Thalassiosirales.

Line 193-194: Why only a highly-divergent DNMT5a origin? Why not divergent DNMT5b, like "D_XP_005778783_2903"?

Line 198: Was this N-terminal laminin B receptor domain lineage specific? Were the other 14 all thalassiosiralean diatoms as well?

Lines 224-225: I think the claim of an "ancestral" DNMT4 in diatoms is not cut-and-dried, especially as almost half of the DNMT4 transcripts are from one order (the Thalassiosirales) and only two from the "early diverging" clades of the Coscinodiscophytina, and is only present in half of the species in one of those coscinodiscophycean genera (*Corethron*).

Line 255: Again, many of the taxa sampled where DNMT1-like transcripts are found are from the Chlamydomonadales, which is a relatively derived order within the Chlorophyceae. These transcripts are largely missing from the early-diverging Prasinophytina.

Line 291: "...could not be resolved (Additional File 1: Fig. S1)"

Line 296-297: Again, I would question the classification of diatoms as "early-diverging eukaryotes"...particularly when represented by a HIGHLY-derived taxon such as *Phaeodactylum tricorutum*.

Figure 2: Double-check the node on your "ochrophyte" label. Should include Chrysophytes and Raphidophytes.

Line 297: qPCR

Discussion

Lines 463-467: This appears to be the driving factor in the interpretation of the evolutionary history of these markers (particularly in diatoms) within this paper, and should probably be explicitly stated early in the Results section.

Line 470-471: To start, supplying taxon names at the terminal branches in the phylogeny might help to identify potential multi-copy gene families

Line 472: If so, then "lineage" must be explicitly defined in this manuscript for interpretation purposes.

Lines 488-490: These are highly derived diatom genera to make such an inference. When would this duplication have occurred if one of the duplicates is not found (or expressed) in earlier-

diverging diatom taxa?

Line 505: Is this generalization referring to photosynthetic microeukaryotes or all microeukaryotes? Genome organization is quite divergent in dinoflagellates compared to other microalgal groups

Line 526: de-italicize "similar"

Line 540: There are multiple DNMT types in *Phaeodactylum*. Does functional redundancy affect the knockout experiments presented here?

Methods

Line 611-612: Why enrich specifically based on TARA OCEANS abundance? That certainly explains the dominance of marine plankton species in the results, but why purposefully introduce bias?

Reviewer #2 (Remarks to the Author):

In this paper, Huguin et al. conducted a comparative genomic study on cytosine methyl transferases by focusing on the SAR lineage. In addition, genome editing was conducted to knock out DNMT5a genes. While I agree that the knowledge and findings would be useful to those who are interested in the diversity of DNMT genes and diatom epigenomes, my overall impression was that they should be shared via a more specialized journal. The comparative genomics part is too descriptive and rather stressful to read through for those who are not very interested in DNMT enzymes. The KO experiment part gives evidence of DNMT5a function, but its effects on TE expression are still partial; the overlap between the mutant strains is limited for unknown reasons; and does not clarify the total landscape of the DNMT enzyme functions in diatom. Some comments follow:

- L151 more than 99% of true positives: True positives cannot be detected by definition. Precision and recall need to be examined.
- L151 eDAF curation: needs explanation.
- L160: In phylogenetic informatics, there is still debate on whether BI performs better for fast-evolving sequences. In general, models matter more than methods.
- L164 high homology: homology is a qualitative concept and cannot be high or low.
- L166 (and after): The wording of DNMT5"s" is confusing as there are DNMT5a, b, and c.
- L166-168: How will this serine substitution be interpreted?
- L193: might also be from DNMT5b.
- L211: Given the sister clade of DNMT4 and DNMT1 is not strongly supported, the discussion in this section is too speculative.
- Ultra-long paragraphs need to be revised to make their logic structures clearer.
- English editing is needed. Just as some examples,
 - Grammar errors (e.g., L186: DNMT5a and "a" DNMT5b gene copies)
 - Wording (e.g., L129: The MMETSP "concatenates", L136 approach with "studies")

Reviewer #3 (Remarks to the Author):

The authors present a phylogenetic study of DNA methyltransferases (DNMTs) in microalgae, supplemented with functional analyses of one DNMT in *P. tricornutum*, showing that this DNMT is central in DNA methylation in this species.

To my knowledge, this is the first in-depth analysis of the distribution and diversity of DNMTs in microeukaryotes, although previous studies (cited in the manuscript) have done more limited analyses. Also, the authors present the first functional study using DNMT knockouts in a diatom. Thus, this study presents novel results that should be of interest within the fields of both epigenetics and diatom/phytoplankton biology.

In general, the experiments appear to be well designed and performed. The language and organisation of the manuscript is also of good quality. I have no major concerns regarding this manuscript. My main minor issue is the lack of details regarding the verification of homozygous disruption of the DNMT5a gene, as commented below.

Abstract:

Line 39: Please add the abbreviation for Stramenopiles-Alveolate-Rhizaria (SAR), as it is being used in the last sentence of the abstract.

Introduction:

Line 70-71: The different description of DNMT3 (family) and DNMT1 (enzymes) is confusing. Is there a reason why the different wording is used?

Results

Line 222-224: This sentence is a bit confusing with regard to the numbers presented and their connection to centric and pennate species. Please reformulate.

Line 225: With "other species", are the authors referring to other species than diatoms?

Line 236: Typo: Chattonella subsalsa.

Line 317 and throughout the manuscript: The Phaeodactylum reference line is named in various ways, none of which are correct. The correct name is Pt1 8.6.

Line 384-387: This sentence is a bit unclear – please rephrase.

Discussion:

Line 534-535: I have problems understanding where the numbers 51/9,416 come from and what they indicate.

Materials and methods:

Line 659: I suggest to replace "extinction" with "disruption".

Line 660: Typo: CRISPR

Line 660-679: The section on CRISPR/Cas9-based gene editing contains no information regarding screening for and verification of DNMT5a knockouts. Necessary details should be added besides citing Ref 64., either to this section or as a separate section. Also, there is no information whether disruption of both DNMT5a alleles was verified. The results clearly indicate that all KO lines are loss-of-function; still, this should be clarified.

Figures:

Figure 1: Some of the colours used in the colour legend for protein domains are quite similar, especially the red nuances indicating BAH and LBR domains. Please modify the colouring for improved readability.

Figure 2: The figure indicates that Raphidophytes have no DNMTs, in contrast to the results presented in line 234-239. I also note that the column for "DNMT1-like" is empty...

Supplementary Information:

Fig S1: Please modify the colour coding for protein domains similar as in Fig. 1. Also add the colour legend that is used in Fig. 1.

Fig. S3: Subfigure a should show sequence data for both alleles to convincingly prove that the M23 and M25 lines are homozygous DNMT5a mutations.

Table S15: please add primer sequences for the housekeeping genes (RPS and TUB)

References:

A majority of the references lack volume number and/or page numbers/article number.

We would like to thank the reviewers for taking the time to assess our manuscript and for their valuable comments and suggestions. We have carefully considered their comments and below a point by point response to all reviewers' concerns with tracked changes in the text manuscript.

Reviewers' comments:

Reviewer #1 (Remarks to the Author):

I appreciate the combined bioinformatic and laboratory approach in this study, investigating the mechanisms of DNA methylation in diatoms. I agree with the authors in the potential importance and reach of this work and find their methodology to be appropriate. The manuscript is largely well-written and comprehensible.

I would only suggest a more nuanced interpretation of the results, particularly in regard to the evolution and phylogenetic interpretation. I've laid out some of my concerns below.

Introduction

Line 144: Do the authors consider diatoms an “early diverging eukaryote lineage”? I don't know that I would consider a phylogenetically-derived product of a secondary endosymbiotic event that diverged some 200 mya as “early diverging” for a eukaryote...

Response:

We thank the reviewer for this remark. The molecular time scale for eukaryote evolution was revisited in Strassert et al., Nature Comm 2021 (<https://doi.org/10.1038/s41467-021-22044-z>), where the origination period for the lineages currently harboring red algal-derived plastids such as diatoms was inferred using Bayesian molecular clock analyses applied on a phylogenomic dataset and was found to be between 1298 and 622 mya. Therefore, we considered diatoms as early diverging eukaryotes.

Results

Figure S1: Taxon identities would be more useful as branch termini to assess this phylogeny than these sequence codes. Are these branches multiple copies of the marker in species, or different species?

Response:

We thank the reviewer for the suggestion. Taxon identities were added to sequence codes. Branches represent different species. It is now clear with species names instead of sequence codes.

>Line 182-184: Explain what you mean by this statement. The topology of the enzyme is unconvincing, or the phylogenetic position of these *E. huxleyi* sequences is suspect because they are not monophyletic?

Response:

The sentence was rephrased for more clarity and now reads as follows: line 188-192: “In addition, the phylogenetic position of *E. huxleyi* DNMT5 proteins are not very supportive. First, *E. huxleyi* DNMT5a protein is not monophyletic with other ochrophyte DNMT5s and its DNMT5b appears as an outgroup found in diatoms”.

Haptophyte DNMT5a and DNMT5b proteins contain classical domains observed in other non-dinoflagellate DNMT5s while DNMT5b is likely from diatom/bolidomonas origin. It is possible that *E. huxleyi* inherited a DNMT5 pair from a common ancestor that then extensively diverged but we could not annotate other DNMT5s with certitudes in the other haptophytes despite partial transcripts resembling SNF-like proteins in *Phaeocystis antarctica* for exemple (Table S2). Our phylogeny is only based on the DNMT domain of DNMT5 and not on the SNF domain; we thus did not include those proteins in the trees.

> Line 186-187: “Lineage specific loss” is a tough sell here when the “lineages” are so broadly defined as “centric versus pennate”. These states are broad morphological categories which do not necessarily correspond to genetic lineages. The “centric” diatoms, for example, are not monophyletic by DNA sequence data and represent several genetic lineages. In fact, among the taxa labeled “centric” in Table S3, almost 65% are from the same taxonomic order—the Thalassiosirales.

Response:

We specified in the text (line 194-195) that the loss indeed occurred in Thalassiosirales based on available sequencing data. Of note, even within raphid and araphid pennate diatoms for which genomes are available, DNMT5a and DNMT5b were likely lost multiple times such as in *Haslea*, *Navicula* and *Pleurosigma* (table S3). Table S3 was modified to distinguish between raphid and araphid pennate as well as polar and radial centrics. Nonetheless, those losses must be interpreted based on transcript abundance within the MMETSP database and poor genomic annotations of diatom genomes. More DNMTs might be found in the future. This is the reason why we do not push the analysis to more specifically account for differences between rapid and araphid pennates.

Line 193-194: Why only a highly-divergent DNMT5a origin? Why not divergent DNMT5b, like “D_XP_005778783_2903”?

Response:

We agree with the reviewer that the evolutionary history of DNMT5s are not well resolved based on this phylogeny. Dinoflagellate DNMT5c could originate from an ancestral DNMT5 that diverged compared to the other DNMT5a/b (absence of SNF domain). The sentence now reads as follows line 198-199 “Dinoflagellate DNMT5c sequences may thus represent a highly divergent DNMT subgroup that our phylogeny failed to associate with other DNMT5s.”

Line 198: Was this N-terminal laminin B receptor domain lineage specific? Were the other 14 all thalassiosiralean diatoms as well?

Response:

Table S3 and its legend were modified so that DNMT5b with LRB domains are indicated. This domain is indeed found in thalassiosirales (such as *Skeletonema costatum* and *marinoi*) but also in the pennate *Fragilariopsis cylindrus* and other centrics such as the naviculale *Amphiprora*_sp. We think this domain is a novelty of DNMT5b compared to other DNMT5s.

Lines 224-225: I think the claim of an “ancestral” DNMT4 in diatoms is not cut-and-dried, especially as almost half of the DNMT4 transcripts are from one order (the Thalassiosirales) and only two from the “early diverging” clades of the Coscinodiscophytina, and is only present in half of the species in one of those coscinodiscophycean genera (*Corethron*).

Response:

We agree that the exact origin of DNMT4 within diatoms is far from clear. The MMETSP database is based on transcriptomes therefore the absence of DNMT4 in other centrics could be due to the culture condition used for each species. As you mentioned later we made the limits of the MMETSP appear earlier in the text. Considering this, we think that the observations of DNMT4 in at least some centrics outside thalassiosirales is sufficient to say that this family was present in the common ancestor of diatoms.

Line 255: Again, many of the taxa sampled where DNMT1-like transcripts are found are from the Chlamydomonadales, which is a relatively derived order within the Chlorophyceae. These transcripts are largely missing from the early-diverging Prasinophytina.

Response:

We agree, and we modified the sentence to remove the mention of plant evolution and to take into account the presence of DNMT1-like sequences in Chlamydomonadales specifically. (line 255-258). In fact, prasinophytes seem to have opted for the DNMT5a family instead.

Line 291: “...could not be resolved (Additional File 1: Fig. S1)”

Response:

This was corrected. (line 293)

Line 296-297: Again, I would question the classification of diatoms as “early-diverging eukaryotes”...particularly when represented by a HIGHLY-derived taxon such as *Phaeodactylum tricornutum*.

Response:

Please see response 1 above

Figure 2: Double-check the node on your “ochrophyte” label. Should include Chrysophytes and Raphidophytes.

Response:

The node was checked and includes now Chrysophytes and Raphidophytes.

Line 297: qPCR

Response:

was corrected line 402

Discussion

Lines 463-467: This is appears to be the driving factor in the interpretation of the evolutionary history of these markers (particularly in diatoms) within this paper, and should probably be explicitly stated early in the Results section.

Response:

We thank the reviewer for the remark. This is now stated early in the results section. Line 164-167: “ The MMETSP database is composed of transcripts and is thus deprived from lowly expressed genes or pseudogenes. The relative absence of a given gene family within a species will therefore be interpreted accordingly.”

Line 470-471: To start, supplying taxon names at the terminal branches in the phylogeny might help to identify potential multi-copy gene families

Response:

We thank the reviewer for the suggestion. Taxon names are now added at the branches of the tree.

Line 472: If so, then “lineage” must be explicitly defined in this manuscript for interpretation purposes.

Response: line 475: the mention lineage is confusing and was removed

Lines 488-490: These are highly derived diatom genera to make such an inference. When would this duplication have occurred if one of the duplicates is not found (or expressed) in earlier-diverging diatom taxa?

Response: We removed from the text manuscript the following “This scenario is supported by the presence of both DNMT5a and b orthologues in the genome of *F. cylindrus* and *Synedra* species”. Based on our data it is hard to speculate on the origin of each DNMT5 within their respective lineages notably due to the lack of radial centric diatom genomes in databases. We also removed the sentence: “This suggests that stramenopiles show an ancestral duplication of DNMT5s, which are differentially retained as DNMT5b or DNMT5a in diatoms and *B. pacifica*.” that could be found line 187-189 of the manuscript.

Line 505: Is this generalization referring to photosynthetic microeukaryotes or all microeukaryotes? Genome organization is quite divergent in dinoflagellates compared to other microalgal groups

Response:

We were referring to the green microalgae presented in Huff and Zilberman et al 2014. It is true that the specific function of DNMT5 in the regulation of highly compacted genome is still to this date a speculation. The sentence now reads as follow line 507-509 “This domain architecture might be due to the compact genomes of the green microalgae *Micromonas pusilla*, *Ostreococcus tauri* and *Bathycoccus prasinos*” and we added the reference.

Line 526: de-italicize “similar”

Response:

done

Line 540: There are multiple DNMT types in *Phaeodactylum*. Does functional redundancy affect the knockout experiments presented here?

Response:

The expression level of other dnmts (DNMT3, DNMT6) is not different in the DNMT5 KO and wild type. They are also predicted *de novo* methyltransferases rather than maintenance. We did not find convincing compensation at the methylation level in absence of DNMT5.

Methods

Line 611-612: Why enrich specifically based on TARA OCEANS abundance? That certainly explains the dominance of marine plankton species in the results, but why purposefully introduce bias?

Response

We didn't aim at enriching species from marine environments. The only purpose was to enrich the reference database with more species and because there are not that many genomes/transcriptomes available, we included the available sequenced species that happen to be marine.

Reviewer #2 (Remarks to the Author):

In this paper, Hogue et al. conducted a comparative genomic study on cytosine methyl transferases by focusing on the SAR lineage. In addition, genome editing was conducted to knock out DNMT5a genes. While I agree that the knowledge and findings would be useful to those who are interested in the diversity of DNMT genes and diatom epigenomes, my overall impression was that they should be shared via a more specialized journal. The comparative genomics part is too descriptive and rather stressful to read through for those who are not very interested in DNMT enzymes. The KO experiment part gives evidence of DNMT5a function, but its effects on TE expression are still partial; the overlap between the mutant strains is limited for unknown reasons; and does not clarify the total landscape of the DNMT enzyme functions in diatom. Some comments follow:

- L151 more than 99% of true positives: True positives cannot be detected by definition. Precision and recall need to be examined.

Response: Line 151: The sentence was misphrased. we modified the sentence for more precision. Now it reads as follows line 152: "The aim of this approach was to retrieve the maximum of DNMT hits between distantly related eukaryotes"

- L151 eDAF curation: needs explanation.

Response: The term "curation" was confusing. We rephrased the sentence as follow : "We retained sequences showing conserved DNMT domains and depicted their domain structures using eDAF analysis"

- L160: In phylogenetic informatics, there is still debate on whether BI performs better for fast-evolving sequences. In general, models matter more than methods.

Response:

The tree construction exploited the stability of Bayesian approaches to deal with the fast evolution rates observed in our DNMT sequences. Methods based on posterior probabilities present more stable support values than random sampling algorithms when facing high mutation rates.

- L164 high homology: homology is a qualitative concept and cannot be high or low.

Response: The term “high” was removed line 170

- L166 (and after): The wording of DNMT5"s" is confusing as there are DNMT5a, b, and c.

Response:

We thank the referee for this remark. We specified whenever appropriate whether it is all dnmt5 or one of the three. (i.e DNMT5a, b and c)

- L166-168: How will this serine substitution be interpreted?

Response:

We thank the reviewer for this very interesting comment. Recent studies demonstrated how DNMT5a catalyzes the methylation of DNA and this requires the ATPase activity of its SNF domain. *In vitro* experiments indeed first showed that the interconnection between the SNF domain (ATPase) and the DNA binding motifs recognizing hemimethylated DNA of DNMT5a of the yeast *Cryptococcus neoformans* is strictly required for the function of the enzyme (ref1). Crystallography experiments (ref2) further demonstrated that the Serine substitution, along with other substitutions found in motifs I and V, generates an hydrogen bound network preventing SAM binding. This is the hydrolysis of ATP by the SNF domain that places the protein into a fully functional conformation. In absence of hemi-methylated DNA recognition the enzyme is locked in an auto-inhibitory state.

In our study, we confirm that these amino acid substitutions are conserved in DNMT5a, DNMT5b and DNMT5c enzymes. We modified the supplementary figure 2 to indicate those amino acid positions and mentioned their role line 173-176

ref1: Dumesic, P. A., Stoddard, C. I., Catania, S., Narlikar, G. J. & Madhani, H. D. ATP Hydrolysis by the SNF2 Domain of Dnmt5 Is Coupled to Both Specific Recognition and Modification of Hemimethylated DNA. *Mol. Cell* (2020) doi:10.1016/j.molcel.2020.04.029.

ref2: Wang, J. et al. Structural insights into DNMT5-mediated ATP-dependent high-fidelity epigenome maintenance. *Mol. Cell* 82, 1186-1198.e6 (2022).

- L193: might also be from DNMT5b.

Response: We agree with the reviewer that the evolutionary history of DNMT5s are not well resolved based on this phylogeny. Dinoflagellate DNMT5c could originate from an ancestral DNMT5 that diverged compared to the other DNMT5a/b. The sentence now reads as follows line 198 "Dinoflagellate DNMT5c sequences may thus represent a highly divergent DNMT subgroup that our phylogeny failed to associate with other DNMT5s."

- L211: Given the sister clade of DNMT4 and DNMT1 is not strongly supported, the discussion in this section is too speculative.

Response:

We agree with the reviewer that the relationship between DNMT4 and DNMT1 is poorly supported as we mentioned in the paper line 212. As such, the section was renamed "The DNMT4 and DNMT1 family methyltransferases in microalgae" and the following sentence was removed "Together, these data rather suggest that diatoms, fungi and raphidophyceae enzymes are paraphyletic DNMT1-divergent gene families" to make the section more focused on the description of the data.

- Ultra-long paragraphs need to be revised to make their logic structures clearer.

- English editing is needed. Just as some examples,

-- Grammar errors (e.g., L186: DNMT5a and "a" DNMT5b gene copies)

-- Wording (e.g., L129: The MMETSP "concatenates", L136 approach with "studies")

Response:

The paragraphs were shortened and carefully edited.

Reviewer #3 (Remarks to the Author):

The authors present a phylogenetic study of DNA methyltransferases (DNMTs) in microalgae, supplemented with functional analyses of one DNMT in *P. tricornutum*, showing that this DNMT is central in DNA methylation in this species.

To my knowledge, this is the first in-depth analysis of the distribution and diversity of DNMTs in microeukaryotes, although previous studies (cited in the manuscript) have done more limited analyses. Also, the authors present the first functional study using DNMT knockouts in a diatom. Thus, this study presents novel results that should be of interest within the fields of both epigenetics and diatom/phytoplankton biology.

In general, the experiments appear to be well designed and performed. The language and organisation of the manuscript is also of good quality. I have no major concerns regarding this manuscript. My main minor issue is the lack of details regarding the verification of homozygous disruption of the DNMT5a gene, as commented below.

Abstract:

Line 39: Please add the abbreviation for Stramenopiles-Alveolate-Rhizaria (SAR), as it is being used in the last sentence of the abstract.

Response: The abbreviation was added in the abstract

Introduction:

Line 71-72: The different description of DNMT3 (family) and DNMT1 (enzymes) is confusing. Is there a reason why the different wording is used?

Response:

The word 'enzymes' was replaced by 'families' line 72-73.

DNMTs are a family of proteins with enzymatic activity; we thus use the term enzymes/proteins/family without distinctions throughout the manuscript.

Results

Line 222-224: This sentence is a bit confusing with regard to the numbers presented and their connection to centric and pennate species. Please reformulate.

Response:

We rephrased the sentence as follows: "A total of 8 pennate diatoms and 23 centric diatoms out of 60 species, express or encode at least one DNMT4 related transcript (Additional File 2: Table S3)" line 227-229.

Line 225: With "other species", are the authors referring to other species than diatoms?

Response:

We specified "of the MMETSP database" line 231

Line 236: Typo: Chattonella subsalsa.

Response:

Was corrected. Was also corrected in Tables and figures

Line 317 and throughout the manuscript: The *Phaeodactylum* reference line is named in various ways, none of which are correct. The correct name is Pt1 8.6.

Response:

We thank the reviewer for the remark. The correct name is now used throughout the text

Line 384-387: This sentence is a bit unclear – please rephrase.

Response:

The sentence was rephrased and reads as follows: "In contrast, only 219 (16%) of protein coding genes are overexpressed in both mutants. They show clear enrichment for GOs associated with protein folding as well as nucleotide phosphate metabolism and nucleotide binding activity (Fig. 4e, Additional File 2: Table S11)" line 389-393.

Discussion:

Line 534-535: I have problems understanding where the numbers 51/9,416 come from and what they indicate.

Response:

We specified as follow: "Only 51 out of all 9,416 annotated genes are found within DMRs" line 537

Materials and methods:

Line 659: I suggest to replace "extinction" with "disruption".

Response:

Extinction was replaced by disruption

Line 662: Typo: CRISPR

Response:

Was corrected

Line 660-679: The section on CRISPR/Cas9-based gene editing contains no information regarding screening for and verification of DNMT5a knockouts. Necessary details should be added besides citing Ref 64., either to this section or as a separate section. Also, there is no

information whether disruption of both DNMT5a alleles was verified. The results clearly indicate that all KO lines are loss-of-function; still, this should be clarified.

Response:

We added more details on how mutants were selected in the method section.

The following response is associated with a new supplementary figure S3 that we wish to add to the paper. The panel b, c and d of the original supplementary figure S3 have been moved to a new supplementary figure 4. The annotation of the other supplementary figures were adjusted to account for these modifications.

Our original primer design allowing the sequencing of the mutations is not allele specific. If mutants were heterozygotes we should have seen a mixture of sequences after Sanger sequencing. We observed this during selection of clones for which it was shown that they were mutant/WT mixture/contaminations (data not shown).

We nonetheless reasoned that we could amplify a region with the mutation present in M23 / M25 in the first exon along with hypothetical variations present in the region targeted by Cas9. We thus designed a new primer pair named DNMT5locus_F and R (table S15) that amplifies a 960bp region that includes the first exon (Cas9 target sites) and the promoter region of DNMT5 (Figure S3). The PCR product was purified and sequenced using the same F and R primers. As expected, the sequence from the WT region indicates a set of polymorphisms (including putative deletions and multiple insertions) that all locate to the promoter region of DNMT5. This is seen by the mixture of sanger sequencing traces for the WT. Using the reverse primer (Figure S3a), we found that the mutations in M23/M25 are homozygous. However, mutants are also homozygous for the other polymorphisms identified in the WT. To be precise, the WT traces are a mixture of the alleles found in M23 and M25. This is also observed using the forward sequencing primer that further resolves single nucleotide polymorphisms in the promoter region in 5' of a poly C repeat hindering sequencing (Figure S3b). We believe that M23 and M25 only amplify one of the two possible alleles present in the region. This is not the first time that homozygous mutants are primarily observed in diatoms after Cas9 double strand breaks (ref1) as we found a loss of heterozygosity, this is evocative of gene conversion.

ref1: 1. Nymark, M., Sharma, A. K., Sparstad, T., Bones, A. M. & Winge, P. A CRISPR/Cas9 system adapted for gene editing in marine algae. *Sci. Rep.* (2016) doi:10.1038/srep24951.

Figures:

Figure 1: Some of the colours used in the colour legend for protein domains are quite similar, especially the red nuances indicating BAH and LBR domains. Please modify the colouring for improved readability.

Response:

The colors were made different

Figure 2: The figure indicates that Raphidophytes have no DNMTs, in contrast to the results presented in line 234-239. I also note that the column for “DNMT1-like” is empty...

Response: We thank the reviewer for the comment, The DNMT1-like enzymes of raphidophytes were indeed not indicated. The figure was modified accordingly

Supplementary Information:

Fig S1: Please modify the colour coding for protein domains similar as in Fig. 1. Also add the colour legend that is used in Fig. 1.

Response:

The color coding was modified according to Fig. 1.

Fig. S3: Subfigure a should show sequence data for both alleles to convincingly prove that the M23 and M25 lines are homozygous DNMT5a mutations.

Response:

a new supplementary figure S3 was generated (see response above)

Table S15: please add primer sequences for the householding genes (RPS and TUB)

Response:

The primers sequences of RPS and TUB genes were added

References:

A majority of the references lack volume number and/or page numbers/article number.

Response:

Reference list was carefully revised to include the missing information.

On behalf of all the authors,

L. Tirichine

REVIEWERS' COMMENTS:

Reviewer #1 (Remarks to the Author):

I am largely satisfied with how the authors addressed the reviewers' concerns and would encourage the publication of this manuscript.

I would suggest one more edit, however. I appreciate the authors' acknowledgment of the the source of their claim that the diatoms represent an "early-diverging eukaryote lineage" (line 145), but this source should be cited in the text as well.

Reviewer #3 (Remarks to the Author):

I am pleased to see that my comments have been addressed. I have no further comments.

We would like to thank the reviewers for taking the time to assess our manuscript and for their valuable comments and suggestions. We have carefully considered their comments and below a point by point response to all reviewers' concerns with tracked changes in the text manuscript.

Reviewers' comments:

Reviewer #1 (Remarks to the Author):

I appreciate the combined bioinformatic and laboratory approach in this study, investigating the mechanisms of DNA methylation in diatoms. I agree with the authors in the potential importance and reach of this work and find their methodology to be appropriate. The manuscript is largely well-written and comprehensible.

I would only suggest a more nuanced interpretation of the results, particularly in regard to the evolution and phylogenetic interpretation. I've laid out some of my concerns below.

Introduction

Line 144: Do the authors consider diatoms an “early diverging eukaryote lineage”? I don't know that I would consider a phylogenetically-derived product of a secondary endosymbiotic event that diverged some 200 mya as “early diverging” for a eukaryote...

Response:

We thank the reviewer for this remark. The molecular time scale for eukaryote evolution was revisited in Strassert et al., Nature Comm 2021 (<https://doi.org/10.1038/s41467-021-22044-z>), where the origination period for the lineages currently harboring red algal-derived plastids such as diatoms was inferred using Bayesian molecular clock analyses applied on a phylogenomic dataset and was found to be between 1298 and 622 mya. Therefore, we considered diatoms as early diverging eukaryotes.

Results

Figure S1: Taxon identities would be more useful as branch termini to assess this phylogeny than these sequence codes. Are these branches multiple copies of the marker in species, or different species?

Response:

We thank the reviewer for the suggestion. Taxon identities were added to sequence codes. Branches represent different species. It is now clear with species names instead of sequence codes.

>Line 182-184: Explain what you mean by this statement. The topology of the enzyme is unconvincing, or the phylogenetic position of these *E. huxleyi* sequences is suspect because they are not monophyletic?

Response:

The sentence was rephrased for more clarity and now reads as follows: line 188-192: “In addition, the phylogenetic position of *E. huxleyi* DNMT5 proteins are not very supportive. First, *E. huxleyi* DNMT5a protein is not monophyletic with other ochrophyte DNMT5s and its DNMT5b appears as an outgroup found in diatoms”.

Haptophyte DNMT5a and DNMT5b proteins contain classical domains observed in other non-dinoflagellate DNMT5s while DNMT5b is likely from diatom/bolidomonas origin. It is possible that *E. huxleyi* inherited a DNMT5 pair from a common ancestor that then extensively diverged but we could not annotate other DNMT5s with certitudes in the other haptophytes despite partial transcripts resembling SNF-like proteins in *Phaeocystis antarctica* for exemple (Table S2). Our phylogeny is only based on the DNMT domain of DNMT5 and not on the SNF domain; we thus did not include those proteins in the trees.

> Line 186-187: “Lineage specific loss” is a tough sell here when the “lineages” are so broadly defined as “centric versus pennate”. These states are broad morphological categories which do not necessarily correspond to genetic lineages. The “centric” diatoms, for example, are not monophyletic by DNA sequence data and represent several genetic lineages. In fact, among the taxa labeled “centric” in Table S3, almost 65% are from the same taxonomic order—the Thalassiosirales.

Response:

We specified in the text (line 194-195) that the loss indeed occurred in Thalassiosirales based on available sequencing data. Of note, even within raphid and araphid pennate diatoms for which genomes are available, DNMT5a and DNMT5b were likely lost multiple times such as in *Haslea*, *Navicula* and *Pleurosigma* (table S3). Table S3 was modified to distinguish between raphid and araphid pennate as well as polar and radial centrics. Nonetheless, those losses must be interpreted based on transcript abundance within the MMETSP database and poor genomic annotations of diatom genomes. More DNMTs might be found in the future. This is the reason why we do not push the analysis to more specifically account for differences between rapid and araphid pennates.

Line 193-194: Why only a highly-divergent DNMT5a origin? Why not divergent DNMT5b, like “D_XP_005778783_2903”?

Response:

We agree with the reviewer that the evolutionary history of DNMT5s are not well resolved based on this phylogeny. Dinoflagellate DNMT5c could originate from an ancestral DNMT5 that diverged compared to the other DNMT5a/b (absence of SNF domain). The sentence now reads as follows line 198-199 “Dinoflagellate DNMT5c sequences may thus represent a highly divergent DNMT subgroup that our phylogeny failed to associate with other DNMT5s.”

Line 198: Was this N-terminal laminin B receptor domain lineage specific? Were the other 14 all thalassiosiralean diatoms as well?

Response:

Table S3 and its legend were modified so that DNMT5b with LRB domains are indicated. This domain is indeed found in thalassiosirales (such as *Skeletonema costatum* and *marinoi*) but also in the pennate *Fragilariopsis cylindrus* and other centrics such as the naviculale *Amphiprora*_sp. We think this domain is a novelty of DNMT5b compared to other DNMT5s.

Lines 224-225: I think the claim of an “ancestral” DNMT4 in diatoms is not cut-and-dried, especially as almost half of the DNMT4 transcripts are from one order (the Thalassiosirales) and only two from the “early diverging” clades of the Coscinodiscophytina, and is only present in half of the species in one of those coscinodiscophycean genera (*Corethron*).

Response:

We agree that the exact origin of DNMT4 within diatoms is far from clear. The MMETSP database is based on transcriptomes therefore the absence of DNMT4 in other centrics could be due to the culture condition used for each species. As you mentioned later we made the limits of the MMETSP appear earlier in the text. Considering this, we think that the observations of DNMT4 in at least some centrics outside thalassiosirales is sufficient to say that this family was present in the common ancestor of diatoms.

Line 255: Again, many of the taxa sampled where DNMT1-like transcripts are found are from the Chlamydomonadales, which is a relatively derived order within the Chlorophyceae. These transcripts are largely missing from the early-diverging Prasinophytina.

Response:

We agree, and we modified the sentence to remove the mention of plant evolution and to take into account the presence of DNMT1-like sequences in Chlamydomonadales specifically. (line 255-258). In fact, prasinophytes seem to have opted for the DNMT5a family instead.

Line 291: “...could not be resolved (Additional File 1: Fig. S1)”

Response:

This was corrected. (line 293)

Line 296-297: Again, I would question the classification of diatoms as “early-diverging eukaryotes”...particularly when represented by a HIGHLY-derived taxon such as *Phaeodactylum tricornutum*.

Response:

Please see response 1 above

Figure 2: Double-check the node on your “ochrophyte” label. Should include Chrysophytes and Raphidophytes.

Response:

The node was checked and includes now Chrysophytes and Raphidophytes.

Line 297: qPCR

Response:

was corrected line 402

Discussion

Lines 463-467: This is appears to be the driving factor in the interpretation of the evolutionary history of these markers (particularly in diatoms) within this paper, and should probably be explicitly stated early in the Results section.

Response:

We thank the reviewer for the remark. This is now stated early in the results section. Line 164-167: “ The MMETSP database is composed of transcripts and is thus deprived from lowly expressed genes or pseudogenes. The relative absence of a given gene family within a species will therefore be interpreted accordingly.”

Line 470-471: To start, supplying taxon names at the terminal branches in the phylogeny might help to identify potential multi-copy gene families

Response:

We thank the reviewer for the suggestion. Taxon names are now added at the branches of the tree.

Line 472: If so, then “lineage” must be explicitly defined in this manuscript for interpretation purposes.

Response: line 475: the mention lineage is confusing and was removed

Lines 488-490: These are highly derived diatom genera to make such an inference. When would this duplication have occurred if one of the duplicates is not found (or expressed) in earlier-diverging diatom taxa?

Response: We removed from the text manuscript the following “This scenario is supported by the presence of both DNMT5a and b orthologues in the genome of *F. cylindrus* and *Synedra* species”. Based on our data it is hard to speculate on the origin of each DNMT5 within their respective lineages notably due to the lack of radial centric diatom genomes in databases. We also removed the sentence: “This suggests that stramenopiles show an ancestral duplication of DNMT5s, which are differentially retained as DNMT5b or DNMT5a in diatoms and *B. pacifica*.” that could be found line 187-189 of the manuscript.

Line 505: Is this generalization referring to photosynthetic microeukaryotes or all microeukaryotes? Genome organization is quite divergent in dinoflagellates compared to other microalgal groups

Response:

We were referring to the green microalgae presented in Huff and Zilberman et al 2014. It is true that the specific function of DNMT5 in the regulation of highly compacted genome is still to this date a speculation. The sentence now reads as follow line 507-509 “This domain architecture might be due to the compact genomes of the green microalgae *Micromonas pusilla*, *Ostreococcus tauri* and *Bathycoccus prasinos*” and we added the reference.

Line 526: de-italicize “similar”

Response:

done

Line 540: There are multiple DNMT types in *Phaeodactylum*. Does functional redundancy affect the knockout experiments presented here?

Response:

The expression level of other dnmts (DNMT3, DNMT6) is not different in the DNMT5 KO and wild type. They are also predicted *de novo* methyltransferases rather than maintenance. We did not find convincing compensation at the methylation level in absence of DNMT5.

Methods

Line 611-612: Why enrich specifically based on TARA OCEANS abundance? That certainly explains the dominance of marine plankton species in the results, but why purposefully introduce bias?

Response

We didn't aim at enriching species from marine environments. The only purpose was to enrich the reference database with more species and because there are not that many genomes/transcriptomes available, we included the available sequenced species that happen to be marine.

Reviewer #2 (Remarks to the Author):

In this paper, Hogue et al. conducted a comparative genomic study on cytosine methyl transferases by focusing on the SAR lineage. In addition, genome editing was conducted to knock out DNMT5a genes. While I agree that the knowledge and findings would be useful to those who are interested in the diversity of DNMT genes and diatom epigenomes, my overall impression was that they should be shared via a more specialized journal. The comparative genomics part is too descriptive and rather stressful to read through for those who are not very interested in DNMT enzymes. The KO experiment part gives evidence of DNMT5a function, but its effects on TE expression are still partial; the overlap between the mutant strains is limited for unknown reasons; and does not clarify the total landscape of the DNMT enzyme functions in diatom. Some comments follow:

- L151 more than 99% of true positives: True positives cannot be detected by definition. Precision and recall need to be examined.

Response: Line 151: The sentence was misphrased. we modified the sentence for more precision. Now it reads as follows line 152: "The aim of this approach was to retrieve the maximum of DNMT hits between distantly related eukaryotes"

- L151 eDAF curation: needs explanation.

Response: The term "curation" was confusing. We rephrased the sentence as follow : "We retained sequences showing conserved DNMT domains and depicted their domain structures using eDAF analysis"

- L160: In phylogenetic informatics, there is still debate on whether BI performs better for fast-evolving sequences. In general, models matter more than methods.

Response:

The tree construction exploited the stability of Bayesian approaches to deal with the fast evolution rates observed in our DNMT sequences. Methods based on posterior probabilities present more stable support values than random sampling algorithms when facing high mutation rates.

- L164 high homology: homology is a qualitative concept and cannot be high or low.

Response: The term “high” was removed line 170

- L166 (and after): The wording of DNMT5"s" is confusing as there are DNMT5a, b, and c.

Response:

We thank the referee for this remark. We specified whenever appropriate whether it is all dnmt5 or one of the three. (i.e DNMT5a, b and c)

- L166-168: How will this serine substitution be interpreted?

Response:

We thank the reviewer for this very interesting comment. Recent studies demonstrated how DNMT5a catalyzes the methylation of DNA and this requires the ATPase activity of its SNF domain. *In vitro* experiments indeed first showed that the interconnection between the SNF domain (ATPase) and the DNA binding motifs recognizing hemimethylated DNA of DNMT5a of the yeast *Cryptococcus neoformans* is strictly required for the function of the enzyme (ref1). Crystallography experiments (ref2) further demonstrated that the Serine substitution, along with other substitutions found in motifs I and V, generates an hydrogen bound network preventing SAM binding. This is the hydrolysis of ATP by the SNF domain that places the protein into a fully functional conformation. In absence of hemi-methylated DNA recognition the enzyme is locked in an auto-inhibitory state.

In our study, we confirm that these amino acid substitutions are conserved in DNMT5a, DNMT5b and DNMT5c enzymes. We modified the supplementary figure 2 to indicate those amino acid positions and mentioned their role line 173-176

ref1: Dumesic, P. A., Stoddard, C. I., Catania, S., Narlikar, G. J. & Madhani, H. D. ATP Hydrolysis by the SNF2 Domain of Dnmt5 Is Coupled to Both Specific Recognition and Modification of Hemimethylated DNA. *Mol. Cell* (2020) doi:10.1016/j.molcel.2020.04.029.

ref2: Wang, J. et al. Structural insights into DNMT5-mediated ATP-dependent high-fidelity epigenome maintenance. *Mol. Cell* 82, 1186-1198.e6 (2022).

- L193: might also be from DNMT5b.

Response: We agree with the reviewer that the evolutionary history of DNMT5s are not well resolved based on this phylogeny. Dinoflagellate DNMT5c could originate from an ancestral DNMT5 that diverged compared to the other DNMT5a/b. The sentence now reads as follows line 198 "Dinoflagellate DNMT5c sequences may thus represent a highly divergent DNMT subgroup that our phylogeny failed to associate with other DNMT5s."

- L211: Given the sister clade of DNMT4 and DNMT1 is not strongly supported, the discussion in this section is too speculative.

Response:

We agree with the reviewer that the relationship between DNMT4 and DNMT1 is poorly supported as we mentioned in the paper line 212. As such, the section was renamed "The DNMT4 and DNMT1 family methyltransferases in microalgae" and the following sentence was removed "Together, these data rather suggest that diatoms, fungi and raphidophyceae enzymes are paraphyletic DNMT1-divergent gene families" to make the section more focused on the description of the data.

- Ultra-long paragraphs need to be revised to make their logic structures clearer.

- English editing is needed. Just as some examples,

-- Grammar errors (e.g., L186: DNMT5a and "a" DNMT5b gene copies)

-- Wording (e.g., L129: The MMETSP "concatenates", L136 approach with "studies")

Response:

The paragraphs were shortened and carefully edited.

Reviewer #3 (Remarks to the Author):

The authors present a phylogenetic study of DNA methyltransferases (DNMTs) in microalgae, supplemented with functional analyses of one DNMT in *P. tricornutum*, showing that this DNMT is central in DNA methylation in this species.

To my knowledge, this is the first in-depth analysis of the distribution and diversity of DNMTs in microeukaryotes, although previous studies (cited in the manuscript) have done more limited analyses. Also, the authors present the first functional study using DNMT knockouts in a diatom. Thus, this study presents novel results that should be of interest within the fields of both epigenetics and diatom/phytoplankton biology.

In general, the experiments appear to be well designed and performed. The language and organisation of the manuscript is also of good quality. I have no major concerns regarding this manuscript. My main minor issue is the lack of details regarding the verification of homozygous disruption of the DNMT5a gene, as commented below.

Abstract:

Line 39: Please add the abbreviation for Stramenopiles-Alveolate-Rhizaria (SAR), as it is being used in the last sentence of the abstract.

Response: The abbreviation was added in the abstract

Introduction:

Line 71-72: The different description of DNMT3 (family) and DNMT1 (enzymes) is confusing. Is there a reason why the different wording is used?

Response:

The word 'enzymes' was replaced by 'families' line 72-73.

DNMTs are a family of proteins with enzymatic activity; we thus use the term enzymes/proteins/family without distinctions throughout the manuscript.

Results

Line 222-224: This sentence is a bit confusing with regard to the numbers presented and their connection to centric and pennate species. Please reformulate.

Response:

We rephrased the sentence as follows: "A total of 8 pennate diatoms and 23 centric diatoms out of 60 species, express or encode at least one DNMT4 related transcript (Additional File 2: Table S3)" line 227-229.

Line 225: With "other species", are the authors referring to other species than diatoms?

Response:

We specified "of the MMETSP database" line 231

Line 236: Typo: Chattonella subsalsa.

Response:

Was corrected. Was also corrected in Tables and figures

Line 317 and throughout the manuscript: The *Phaeodactylum* reference line is named in various ways, none of which are correct. The correct name is Pt1 8.6.

Response:

We thank the reviewer for the remark. The correct name is now used throughout the text

Line 384-387: This sentence is a bit unclear – please rephrase.

Response:

The sentence was rephrased and reads as follows: "In contrast, only 219 (16%) of protein coding genes are overexpressed in both mutants. They show clear enrichment for GOs associated with protein folding as well as nucleotide phosphate metabolism and nucleotide binding activity (Fig. 4e, Additional File 2: Table S11)" line 389-393.

Discussion:

Line 534-535: I have problems understanding where the numbers 51/9,416 come from and what they indicate.

Response:

We specified as follow: "Only 51 out of all 9,416 annotated genes are found within DMRs" line 537

Materials and methods:

Line 659: I suggest to replace "extinction" with "disruption".

Response:

Extinction was replaced by disruption

Line 662: Typo: CRISPR

Response:

Was corrected

Line 660-679: The section on CRISPR/Cas9-based gene editing contains no information regarding screening for and verification of DNMT5a knockouts. Necessary details should be added besides citing Ref 64., either to this section or as a separate section. Also, there is no

information whether disruption of both DNMT5a alleles was verified. The results clearly indicate that all KO lines are loss-of-function; still, this should be clarified.

Response:

We added more details on how mutants were selected in the method section.

The following response is associated with a new supplementary figure S3 that we wish to add to the paper. The panel b, c and d of the original supplementary figure S3 have been moved to a new supplementary figure 4. The annotation of the other supplementary figures were adjusted to account for these modifications.

Our original primer design allowing the sequencing of the mutations is not allele specific. If mutants were heterozygotes we should have seen a mixture of sequences after Sanger sequencing. We observed this during selection of clones for which it was shown that they were mutant/WT mixture/contaminations (data not shown).

We nonetheless reasoned that we could amplify a region with the mutation present in M23 / M25 in the first exon along with hypothetical variations present in the region targeted by Cas9. We thus designed a new primer pair named DNMT5locus_F and R (table S15) that amplifies a 960bp region that includes the first exon (Cas9 target sites) and the promoter region of DNMT5 (Figure S3). The PCR product was purified and sequenced using the same F and R primers. As expected, the sequence from the WT region indicates a set of polymorphisms (including putative deletions and multiple insertions) that all locate to the promoter region of DNMT5. This is seen by the mixture of sanger sequencing traces for the WT. Using the reverse primer (Figure S3a), we found that the mutations in M23/M25 are homozygous. However, mutants are also homozygous for the other polymorphisms identified in the WT. To be precise, the WT traces are a mixture of the alleles found in M23 and M25. This is also observed using the forward sequencing primer that further resolves single nucleotide polymorphisms in the promoter region in 5' of a poly C repeat hindering sequencing (Figure S3b). We believe that M23 and M25 only amplify one of the two possible alleles present in the region. This is not the first time that homozygous mutants are primarily observed in diatoms after Cas9 double strand breaks (ref1) as we found a loss of heterozygosity, this is evocative of gene conversion.

ref1: 1. Nymark, M., Sharma, A. K., Sparstad, T., Bones, A. M. & Winge, P. A CRISPR/Cas9 system adapted for gene editing in marine algae. *Sci. Rep.* (2016) doi:10.1038/srep24951.

Figures:

Figure 1: Some of the colours used in the colour legend for protein domains are quite similar, especially the red nuances indicating BAH and LBR domains. Please modify the colouring for improved readability.

Response:

The colors were made different

Figure 2: The figure indicates that Raphidophytes have no DNMTs, in contrast to the results presented in line 234-239. I also note that the column for “DNMT1-like” is empty...

Response: We thank the reviewer for the comment, The DNMT1-like enzymes of raphidophytes were indeed not indicated. The figure was modified accordingly

Supplementary Information:

Fig S1: Please modify the colour coding for protein domains similar as in Fig. 1. Also add the colour legend that is used in Fig. 1.

Response:

The color coding was modified according to Fig. 1.

Fig. S3: Subfigure a should show sequence data for both alleles to convincingly prove that the M23 and M25 lines are homozygous DNMT5a mutations.

Response:

a new supplementary figure S3 was generated (see response above)

Table S15: please add primer sequences for the householding genes (RPS and TUB)

Response:

The primers sequences of RPS and TUB genes were added

References:

A majority of the references lack volume number and/or page numbers/article number.

Response:

Reference list was carefully revised to include the missing information.

On behalf of all the authors,

L. Tirichine